# Ampere-hour-scale soft-package potassium-ion hybrid capacitors enabling 6-minute fast-charging

Huanxin Li ®[1,2,3,4,12], Yi Gong[5,6,12], Haihui Zhou[1] ✉, Jing Li[7], Kai Yang[6], Boyang Mao[2], Jincan Zhang ®[2], Yan Shi[8], Jinhai Deng[9], Mingxuan Mao ®[10], Zhongyuan Huang[1], Shuqiang Jiao ®[11] ✉, Yafei Kuang ®[1], Yunlong Zhao ®[5] ✉ & Shenglian Luo[1] ✉

Extreme fast charging of Ampere-hour (Ah)-scale electrochemical energy storage devices targeting charging times of less than 10 minutes are desired to increase widespread adoption. However, this metric is difficult to achieve in conventional Li-ion batteries due to their inherent reaction mechanism and safety hazards at high current densities. In this work, we report 1 Ah soft-package potassium-ion hybrid supercapacitors (PIHCs), which combine the merits of high-energy density of battery-type negative electrodes and high-power density of capacitor-type positive electrodes. The PIHC consists of a defect-rich, high specific surface area N-doped carbon nanotube-based positive electrode, MnO quantum dots inlaid spacing-expanded carbon nanotube-based negative electrode, carbonate-based non-aqueous electrolyte, and a binder- and current collector-free cell design. Through the optimization of the cell configuration, electrodes, and electrolyte, the full cells (1 Ah) exhibit a cell voltage up to 4.8 V, high full-cell level specific energy of 140 Wh kg$^{-1}$ (based on the whole mass of device) with a full charge of 6 minutes. An 88% capacity retention after 200 cycles at 10 C (10 A) and a voltage retention of 99% at 25 ± 1 °C are also demonstrated.

To alleviate the energy crisis and environmental problems caused by the excessive utilization of oil and natural gas, renewable energy and efficient energy storage devices are urgently demanded. Over the past decade, lithium-ion batteries (LIBs) have dominated the market as the most common energy storage device in numerous mobile electronics and electric vehicles. Nevertheless, the shortage of lithium sources limits their applications in large-scale stationary energy storage and

also brings uncertainty to the explosive growth of electric vehicles. On the other hand, the state-of-the-art charging capabilities of LIBs are still far from offering consumers the same refueling experience as conventional vehicles, which has also been considered a critical challenge to ensuring mass-market adoption of electric vehicles[1,2]. Restricted by the intrinsic "rocking-chair" reaction mechanism, it is very difficult to make breakthroughs in the extreme fast charging technologies with

[1]State Key Laboratory for Chemo/Biosensing and Chemometrics, College of Chemistry and Chemical Engineering, Hunan University, Changsha, Hunan 410082, China. [2]Department of Engineering, University of Cambridge, 9 JJ Thomson Avenue, Cambridge CB3 0FA, UK. [3]Department of Chemistry, Physical & Theoretical Chemistry Laboratory, University of Oxford, South Parks Road, Oxford OX1 3QZ, UK. [4]Electrochemical Innovation Lab, Department of Chemical Engineering, University College London, London WC1E 7JE, UK. [5]Dyson School of Design Engineering, Imperial College London, London SW7 2BX, UK. [6]Advanced Technology Institute, University of Surrey, Guildford, Surrey GU2 7XH, UK. [7]Department of Chemical & Process Engineering, University of Surrey, Guildford GU2 7XH, UK. [8]College of Materials and Metallurgy, Guizhou University, Guiyang 550025, China. [9]King's College London, London SE1 1UL, UK. [10]Department of Electrical and Electronic Engineering, Imperial College London, London SW7 2AZ, UK. [11]State Key Laboratory of Advanced Metallurgy, University of Science and Technology Beijing, Beijing 100083, China. [12]These authors contributed equally: Huanxin Li, Yi Gong. ✉e-mail: haihuizh@hnu.edu.cn; sjiao@ustb.edu.cn; yunlong.zhao@imperial.ac.uk; sllou@hnu.edu.cn

targeting charging times of less than 10 min, especially for conventional LIBs at ampere-hour (Ah)-scale, which could bring serious safety hazards. Therefore, the investigation of alternate ion batteries based on abundant metals such as Na and K, as well as the development of novel energy storage devices based on fast energy storage mechanisms, is expected to solve the above problems simultaneously[3–5].

As a promising alternative, potassium-ion batteries (KIBs) have attracted increasing attention for their applications in electric traction power supply and large-scale stationary energy storage. In addition to the abundance and low cost, KIBs have several unique advantages[6]. For example, because the standard redox potential of potassium ($-2.93$ V vs. $E_O$) is close to that of lithium ($-3.04$ V vs. $E_O$), KIBs can deliver high energy density[7]. Since the standard electrode potential of the K/K$^+$ electrode in carbonate ester electrolyte is lower than that of Li/Li$^+$ electrode, the metal deposition on the anode is less likely to occur[8]. The small Stokes radius of potassium ions enables potassium ion-based electrolytes to exhibit high ionic conductivity, thus showing the potential to achieve high-rate performance[7]. Nevertheless, many difficulties remain in KIBs due to the larger radius of potassium ions, such as the design of cathode materials, large volume expansion of electrodes during the charge and discharge process, and the fabrication of full cells. The "rocking chair" mechanism also exacerbates the sluggish intercalation process of potassium ions[9], making it hard to meet the ever-increasing power demand.

To address these challenges, potassium-ion hybrid supercapacitors (PIHCs) were proposed recently, which combine the merits of the high-energy density of KIBs-type anode and the high-power density of capacitor-type cathode[10–12]. This means that PIHCs could overcome the limitations associated with KIB positive electrode materials and rocking-chair reactions while retaining most of the unique advantages of KIBs. Many promising results are demonstrated

at the laboratory scale using coin cell-type PIHCs[13–15]. Pouch cell-type PIHCs have received increased attention lately, and the formation protocol has been optimized for further enhancement of their performance in addition to the improved energy density and cycling stability[16,17]. Nonetheless, their commercialization prospects, especially the practical application in the pouch or cylindrical cells, are still limited. One of the main reasons is the mismatch in capacity and kinetics between cathode and anode materials (Fig. 1a): *a*. the capacitance of the cathode is insufficient since capacity is confined to the supercapacitor level, although rapid kinetics is achievable (Fig. 1a.i); *b*. the layer spacing of conventional graphite anode (~0.34 nm) is relatively narrow for the intercalation and diffusion of potassium ions, which hindered the rapid kinetics (Fig. 1a.iv). Furthermore, conventional PIHC electrolytes tend to cause corrosion of the current collector and side reactions with binders at potentials over 4 V (vs. K$^+$/K), which is another significant limitation for large, stacked pouch cells or cylindrical cells[18].

To address the above issues and push the practical applications of PIHCs to ampere-hour-scale, we propose a holistic design and optimization strategy from three key cell parameters: electrode materials, cell configuration and electrolytes. First, to balance the capacity and kinetics mismatch of capacitor-type cathode and battery-type anode, we propose a strategy of simultaneously modulating surface redox kinetics at the cathode (Fig. 1a.ii) and enhancing ion intercalation at the anode (Fig. 1a.iii), enabling both to approach the pseudocapacitive behavior − continuous and fast reversible redox reactions or intercalation at/near the surface of electrode materials. To prove the concept, we fabricate an ordered nitrogen-doped carbon nanotube array on mesoporous carbon (N-CNTs@MC) with a large specific surface area (1806.3 cm$^2$ g$^{-1}$) as the positive electrode. The optimized porosity and structures can enhance the capacity and rate performance, while

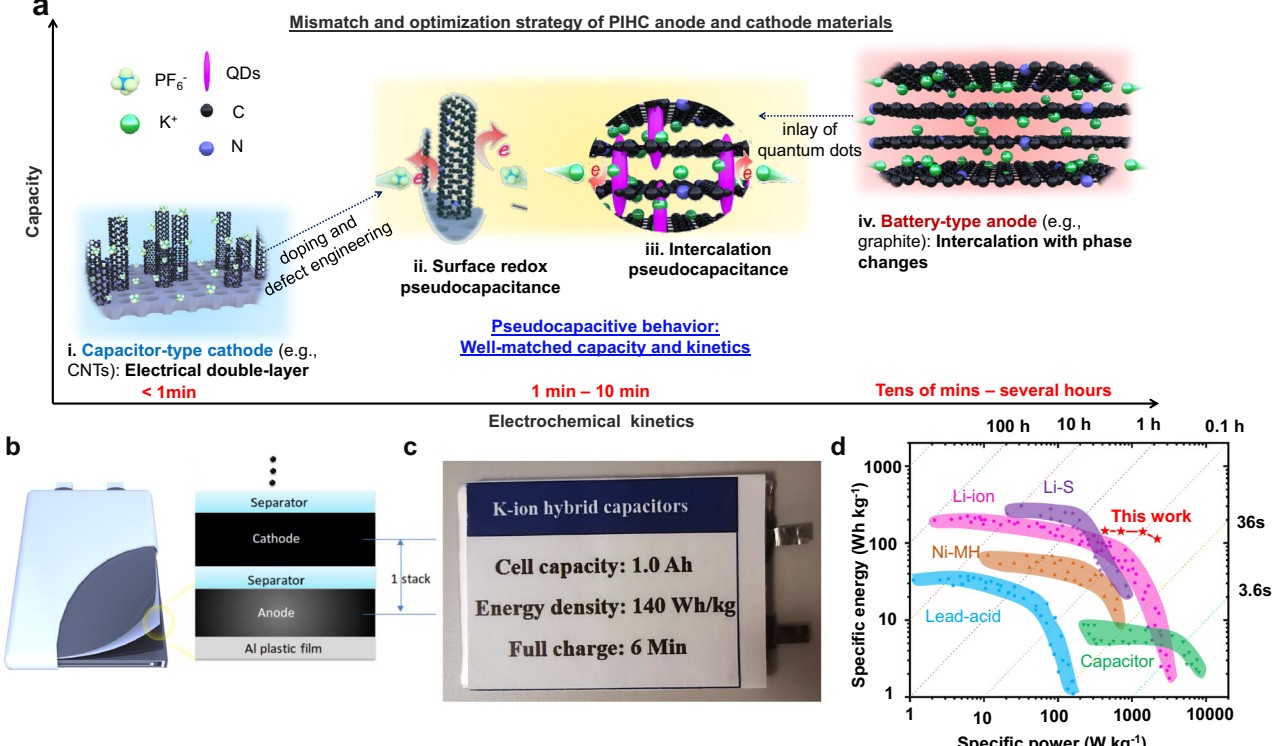

**Fig. 1 | Holistic design and optimization strategy of potassium-ion hybrid supercapacitors (PIHCs) and comparison with commercial energy storage cells. a** The capacity and kinetics mismatch of conventional PIHCs with the capacitor-type cathode (i) and battery-type anode (iv), and the proposed pseudo-capacitive strategy (ii, iii). **b** A scheme of current collector-free and binder-free cell design for soft-package PIHCs. **c** A photograph of fabricated 1 Ah soft-package PIHC. **d** A comparison between the specific energies, powers, and charging rates of the fabricated 1 Ah soft-package PIHCs and the commercial soft-package capacitors and batteries previously reported[42–44].

the surface redox pseudocapacitance introduced by N-doping or defect engineering enables better compatibility with the negative electrode. For the negative electrode, the MnO quantum dots inlaid carbon nanotubes (MnO@CNTs) were synthesized as the electrode. The MnO quantum dots inlaid in the CNT walls can enlarge the interlay space of carbon layers, which can substantially enhance the intercalation kinetics to improve the rate performance[19,20]. In addition, the introduction of MnO can provide enhanced pseudocapacitance at the same time to further improve the whole kinetics, and enable better compatibility with the positive electrode. The combination of the matched cathode and anode can achieve high capacity and high-rate performance simultaneously. Second, to eliminate the side reactions between electrolyte and binder and the corrosions of metal current collectors, a current collector- and binder-free design is demonstrated for both cathode and anode electrodes (Fig. 1b). By mixing with a small fraction of reduced graphene oxide (rGO), the electrode materials could be directly pressed into flexible carbon paper. It's worth mentioning that these free-standing and lightweight electrodes can not only exhibit equivalent sheet conductivity compared to Cu and Al foil, maintaining good rate performance but also reduce the fraction of non-active materials, maximizing the energy density for the stacked pouch cells. In addition, the parameters of the electrolyte, such as the electrolyte weight to cell capacity (the E/C ratio) and electrolyte density, are also optimized. Although a holistic optimization was achieved in the system, enabling the device to deliver specific energy comparable to K ion battery, the system is classified as a potassium hybrid supercapacitor due to its energy storage mechanism[12,13]. With these holistic designs and optimization, our fabricated 1 Ah soft-package PIHCs (Fig. 1c) could exhibit a high voltage of up to 4.8 V and specific energy of 140 W h kg$^{-1}$ based on the whole weight of PIHC pouch cell under a rapid charging at 10 C (equivalent to 6 min), which is a breakthrough in extreme fast charging of Ah-scale energy storage devices (Fig. 1d).

## Results and discussion

### Fabrication and characterization of N-CNTs@MC positive electrode

To obtain the N-CNTs@MC, as illustrated in Supplementary Fig. 2, the precursor Fe$_2$O$_3$-(N-CNTs)@MC was first synthesized by an iron-catalyzed biomass carbon precursor growth approach, after which the Fe$_2$O$_3$ particles would be etched to form hollow CNTs (Fig. 2a) (see Methods). For the comparison, the Fe$_2$O$_3$@MC was prepared without the addition of melamine and etching, while the MC was prepared by annealing the pre-treated carbon powder (MC precursor). SEM and TEM images show Fe$_2$O$_3$ spherical particles were connected on the top of carbon nanotubes[21] (Supplementary Fig. 4). The typical TEM and elemental mapping images of Fe$_2$O$_3$-(N-CNTs)@MC suggest that C and N elements were uniformly distributed on the CNTs while the Fe and O elements were concentrated on the spherical particles(Supplementary Fig. 4b–e). After acid-washing and heat treatment at 3000 °C under a nitrogen atmosphere, the N-CNTs@MC composite was obtained (Supplementary Fig. 5). The nitrogen atmosphere was selected due to the possibility of chemical reactions, reduced contamination, and cost-effectiveness in this process. The SEM image of nitrogen-doped carbon nanotube reveals the nanotube array exhibiting a discernible level of structural organization. The nanotubes display a degree of orderliness and The individual nanotubes are discernible and appear uniformly distributed. XRD pattern of N-CNTs@MC (Supplementary Fig. 9a) shows two sharp peaks at ~26.2° and ~54.8° corresponding to the (002) and (004) plane of graphite carbon, which indicates the N-CNTs@MC was well-graphitized after the high-temperature treatment. Meanwhile, the XPS survey (Supplementary Fig. 9b) of N-CNTs@MC demonstrates that only C (95.7 at.%) and N elements (4.3 at.%) remained. Further, the high-resolution XPS for C 1$s$ (Supplementary Fig. 9c) and N 1s

(Supplementary Fig. 9d) displayed the dominant peaks at 284.6 and 401.0 eV, which reflect the six-ring structural graphitic-C and graphitic-N, respectively[19]. The results reveal that the N-CNTs@MC was in a fine graphitic structure with a small amount of graphitic-N in the graphite plane. BET analysis shows the isotherms of N-CNTs@MC (Supplementary Fig. 11) displayed a closed hysteresis loop of Type-IV, indicating a mesoporous structure. The BET-specific surface area of N-CNTs@MC was calculated to be 1806.3 m$^2$ g$^{-1}$ and the pore size distribution was concentrated on ~5 nm and ~30 nm as well as some micropores (~2 nm). These micropores and ~5 nm pores come from MC, while the pores with a diameter of 30 nm are attributed to the presence of CNTs. The large specific surface area and uniform mesoporous structure of N-CNTs@MC are ascribed to the highly ordered array of CNTs as well as plenty of nano-holes on the CNTs, which is beneficial for the storage and migration of ions during electrochemical processes. To comprehend the porosity, we first conducted a comparison between the performance of N-MC (synthesized using the same method but without Fe catalysts) and N-CNTs@MC (as shown in Supplementary Fig. 12a). This comparison clearly demonstrates that the inclusion of the N-CNTs array markedly enhances the performance of MC. This improvement can be primarily attributed to the increased specific surface and the incorporation of larger nanopores (~30 nm). Additionally, to optimize the porous structure, we examined the performance of N-CNTs@MC fabricated with different ratios of MC precursor and melamine which affect the portions of nanopores (Supplementary Fig. 12b). The result shows that the 1:2 sample exhibits the best performance among these samples. By mixing with a small fraction of rGO (10 ± 0.05 wt%), the N-CNTs@MC can be assembled into free-standing carbon papers which can be folded into different shapes (Fig. 2b). It is observed that the as-fabricated N-CNTs @MC-rGO film consists of rGO flakes and N-CNTs array (Fig. 2a.ii) which are open-ended (Fig. 2c), showing a higher specific surface area. The electronic conductivity of N-CNTs@MC-rGO film is (3.17 ± 0.01) × 10 ^ 5 S m$^{-1}$ at 300 ± 2 K, which is comparable to the commercial Cu foil (Supplementary Fig. 18).

To verify the electrochemical performance of the as-prepared N-CNTs@MC, the half-PIHCs were assembled in coin cell format with the N-CNTs@MC film disk as the positive electrode and a potassium metal disk as the counter and reference electrode. All the tests were conducted within a voltage window of 3.0–4.8 V, where the PF$_6^-$ adsorption and K$^+$ deposition occurred on the N-CNTs@MC and K electrode respectively during the charging process. The CV curves, which were conducted at scan rates of 10, 20, 50, 100 and 200 mV s$^{-1}$, showed an asymmetrical quadrilateral shape (Fig. 2d), indicating the presence of pseudocapacitance in the energy storage mechanism[22]. The double-layer capacitive (DLC) current contribution can be plotted to distinguish the pseudocapacitive contribution. The DLC–pseudocapacitance contribution at the scan rate of 2 mV s$^{-1}$ was calculated and shown in Fig. 2e, and the pseudocapacitance contribution is about 42% which can improve the overall capacitance and the voltage retention ability. The enhanced pseudocapacitive property, which is derived from structural optimization and surface modification (N-doping and defects), notably increases the capacity while maintaining a high-rate performance. The EIS plots of half-PIHCs with N-CNTs@MC and MC as electrodes were compared (Fig. 2f), which demonstrated that the N-CNTs@MC (~21 Ω) has a faster charge transfer than MC (~32 Ω). The corresponding relationship between Z' and $\omega^{-1/2}$ (Fig. 2g) also suggests that the diffusion coefficient of PF$_6^-$ ions in CNNT@MC is larger than that of MC, due to the highly ordered pore structure. The half-PIHCs with N-CNTs@MC positive electrode displayed capacities of ~122.1, 104.8, 96.2, 90.3, 82.5, 75.4, 67.9 mAh g$^{-1}$ at specific currents of 1, 2, 4, 6, 10, 15 and 20 A g$^{-1}$ (Fig. 2h), indicating an excellent rate performance. Even at a high specific current of 5 A g$^{-1}$ (~41.67 C), the half-PIHCs with N-CNTs@MC positive electrode delivered a specific capacity of 92.8 mA h g$^{-1}$ after 500 cycles (Fig. 2i).

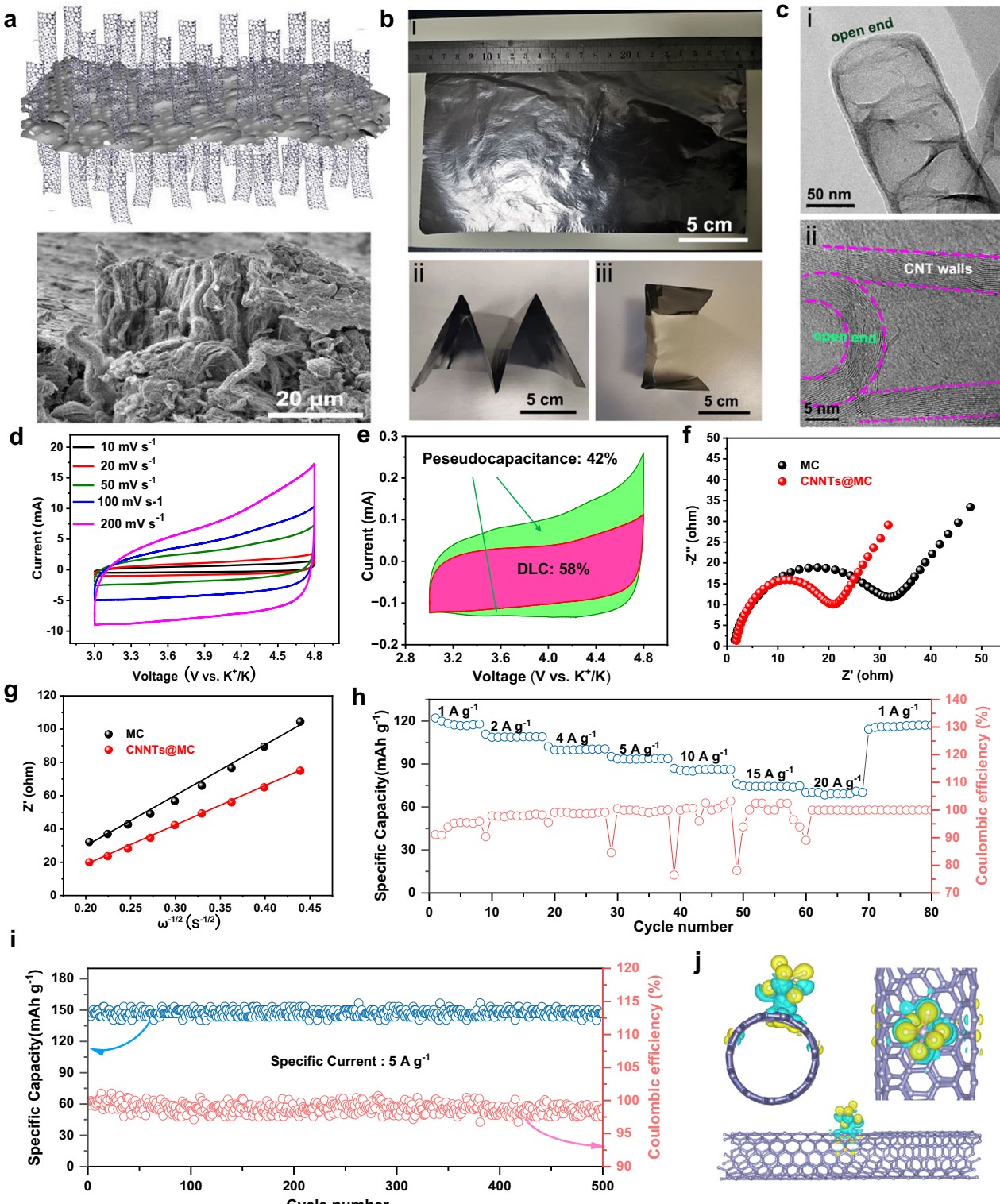

**Fig. 2 | Morphological and electrochemical performance of ordered nitrogen-doped carbon nanotube array on mesoporous carbon (N-CNTs@MC) positive electrode. a** The schematic of the structure of N-CNTs@MC (i) and SEM image of the N-CNTs@MC film (ii). **b** Optical images of pressed flexible N-CNTs@MC films (i) with different sizes and shapes (ii, iii). **c** The morphology characterization of CNTs of the N-CNTs@MC: the TEM image (i) and the enlarged TEM image (ii) of a random CNT. **d** CV curves at scan rates of 10, 20, 50, 100, and 200 mV s⁻¹. **e** The CV curve at 2 mV s⁻¹ and the relative contributions of pseudocapacitance (green area) and double layer capacitance (red area). **f** EIS plots of potassium-ion based half-cell with N-CNT@MC (red) and MC positive electrodes (black) **g** The corresponding relationship between Z' and ω⁻¹/² for N-CNT@MC (red) and MC positive electrodes (black). **h** Rate and CE performance of N-CNT@MC positive electrode in potassium-ion based half-cell. **i.** The cycle stability of N-CNT@MC positive electrode in potassium-ion based half-cell. **j** Charge-Diff of PF₆⁻ ion on N-CNTs based on DFT calculations.

Moreover, the n-type doping of N atoms in carbon nanotubes, not only brings more electron carriers into the material but also alters the distributions of energy bands and surface electrons, which is beneficial for the electron transfer and induction of the pseudo-capacitance via the enhancement of surface redox reactions. DFT calculations were conducted to further understand the adsorption behavior and redox mechanism of $PF_6^-$ ions on CNTs and N-doped CNTs. The front, top, and side views of differential charge density of $PF_6^-$ on CNTs were shown in Fig. 2j, which demonstrated a bond formation and surface electron redistribution after the adsorption, revealing the $PF_6^-$ ions have a strong adsorption ability (−2.19 eV) on CNTs. The $PF_6^-$ adsorption on different locations of CNTs and N-doped CNTs was shown in Supplementary Fig. 13, which revealed that the $PF_6^-$ ion could be adsorbed on both the inner and outer sides of CNTs and N-doped CNTs. The adsorption energies for $PF_6^-$ ions were illustrated in Supplementary Table 2, indicating that the N-doped CNTs possessed a higher adsorption ability (−10.21 eV) to $PF_6^-$ than pure CNTs. To the best of our knowledge, the N-dopped structures in CNTs would alter the electron cloud density of carbon atoms nearby, producing positively charged carbon centers, which is beneficial for the adsorption of $PF_6^-$ ions. Owing to the higher adsorption of anions in the N-CNTs@MC positive electrode, the fast charge/discharge processes were realized.

## Fabrication and characterization of MnO@CNTs negative electrode

Graphite is one of the most popular anode materials for potassium-ion-based devices due to its low cost and super stability, but it suffers from a low specific capacity (~273 mA h g$^{-1}$) and sluggish kinetics resulting from narrow layer spacing[23]. To increase the kinetics of the anode material, the MnO quantum dots well-inlaid carbon nanotubes (MnO@CNTs) were developed. The schematic of synthesis approaches and the structural changes of CNTs are shown in Fig. 3a (see Methods). The detailed morphology and structure of MnO@CNTs are schematically illustrated in Supplementary Fig. 7a, where the manganese oxides are uniformly embedded in the form of quantum dots between the graphite carbon layers, expanding the carbon layer spacing to -0.41 nm. The uniform nanotubes with a hollow structure inside of MnO@CNTs are shown in Supplementary Fig. 7b, c. Further enlargement of the carbon nanotube wall region shows distributed small dark spots (Supplementary Fig. 7d, Fig. 3b.ii) and the atomic resolution image (Supplementary Fig. 7e) shows the obvious atomic structure of MnO crystal, indicating that manganese oxide is present in the carbon nanocomposite in the form of quantum dots. Supplementary Fig. 7f, g are their high-resolution transmission map and the related carbon layer spacing data. The results show that the average carbon layer spacing of the composite is 0.413 nm, which is much larger than the layer spacing of graphite carbon (0.34 nm). The presence of the inlaid MnO is further proved by the XRD measurement (Supplementary Fig. 14a, b), Ramman shifts (Supplementary Fig. 14c), and XPS spectra (Supplementary Fig. 15a). The Raman analysis of MnO@CNTs is presented in Supplementary Fig. 14c. The XRD curve of MnO@CNTs exhibit prominent MnO peaks. When compared to the CNTs, the MnO@CNTs display a broad peak between 24−26 °C, which is attributable to defects resulting from the embedded MnO in the carbon layer and nitrogen doping. Moreover, the peak of MnO@CNTs exhibits a slight leftward shift compared to the CNTs, indicating an enlarged interlayer spacing, consistent with the structural analysis conducted via TEM. The Raman spectrum reveals distinct bands, with the MnO band appearing at 640 cm$^{-1}$, the D band at 1336 cm$^{-1}$, and G band at 1569 cm$^{-1}$, respectively. The D band is attributed to defective edges and lattice distortion, while the G band signifies the crystalline graphitic structure. The intensity ratio of $I_D/I_G$ serves as a measure of carbon defects. In the case of MnO@CNTs, the $I_D/I_G$ ratio is determined to be 1.04, indicating a relatively high level of defects in the MnO@CNTs structure. Compared with the Raman of MWCNTs[24], the broader full

width at half maximum (FWHM) of band G and the absence of the 2D peak at 2696 cm$^{-1}$ further suggest the enhanced defects in the MnO@CNTs. These defects primarily arise from the incorporated MnO (as depicted in Fig. 3h) and some N doping (as shown in Supplementary Fig. 15). The Raman analysis aligns with the findings from XRD and XPS analyses, collectively reinforcing the consistency of the results. The as-obtained MnO@CNTs could be shaped into carbon paper by mixing with a small proportion of rGO (10 wt%) (Fig. 3b), showing good flexibility and good mechanical properties. It is worth mentioning the carbon paper can be pressed with a well-controlled thickness (5−20 µm). Meanwhile, the electronic conductivity of MnO@CNTs film is quite high, making it possible to be used as non-current-collector electrodes for potassium-ion-based devices.

The half-PIHCs with the MnO@CNTs film electrode disk were assembled subsequently to test the corresponding electrochemical performance. Figure 3c shows the CV tests on the half cells at scan rates from 0.1 to 20 mV s$^{-1}$. The reduction peaks correspond to the potassium intercalation process and the shapes of CV curves are well maintained with slight peak shifts at different scan rates, indicating the K ions storage should be dominated by the pseudocapacitive behavior[23]. To verify the contribution of pseudocapacitive behavior, further analysis was conducted as follows. The relationship between the peak current density (*i*) and the sweep rate (*v*) could be described as the following equations:(1) $i = av^b$; (2) $\log(i) = b*\log(v) + \log(a)$; the b value could be determined by the slope of linear fitted log(*i*)-log (*v*) curves[25] (detailed in Supplementary Note 1). Typically, when the b value approaches 0.5, it indicates that the intercalation of potassium ion is mainly determined by the diffusion process, while the b value gets close to 1.0, the potassium storage is dominated by the pseudo-capacitor behaviors, such as surface-controlled capacitive process or intercalation without phase transition[26]. Here the b value in the MnO@CNTs film, as shown in Supplementary Fig. 17, was calculated to be 0.92, which indicates the intercalation of potassium ions tends to display a capacitive behavior after the expanding of interlayer spacing in MnO@CNTs, enabling an excellent rate performance and high capacity. Furthermore, the diffusion and capacitive mixed behaviors can be distinguished by separating the current response at a fixed potential based on the following Eq. (3) $i = k_1v^{1/2} + k_2v$ and (4) $i/v^{1/2} = k_1 + k_2v^{1/2}$, where $k_1$ and $k_2$ are constants, $k_1v^{1/2}$ represents the contribution of diffusion-controlled reaction and $k_2v$ stands for the contribution from the capacitive process[27–29]. According to the fitting curve, the capacitance contribution value and diffusion-control contribution value under each sweep speed are shown in Fig. 3d. The capacitive contributions in MnO@CNTs are 64.8, 70.6, 76.0, 81.2, 86.0, 91.6, 94.5, and 96.6%, at scan rates of 0.1, 0.2, 0.5, 1, 2, 5, 10, 20 mV s$^{-1}$, respectively. The pseudocapacitor-dominated property of MnO@CNTs is well-matched with the N-CNTs@MC as electrodes for PIHCs. To further demonstrate the superiority of MnO@CNTs, the rate and cycling performance of MnO@CNTs disks as negative electrodes for potassium ion half cells were measured in comparison with multi-wall carbon nanotubes (MWCNTs), as shown in Fig. 3f. At the specific currents of 0.2, 0.5, 2, 5, 10, 20 and 40 A g$^{-1}$, the capacities of MnO@CNTs film were 383.3, 338.2, 306.9, 264.6, 233.2, 226.5, and 198.6 mAh g$^{-1}$, while the MWCNTs film potassium ion batteries exhibited capacities of 261.2, 216.5, 172.6, 94.3, 39.8, 7.4, and 1.5 mAh g$^{-1}$, respectively. The capacity and rate performance of MnO@CNTs potassium ion half cells are much better than those of MWCNTs potassium ion cells. Since the interlayer spacing (0.41 nm) of MnO@CNTs composites is much larger than that of MWCNTs (0.34 nm), potassium ions can migrate rapidly between the layers of MnO@CNTs composites. Meanwhile, the inlaid MnO quantum dots promote the surface redox reaction to introduce pseudocapacitance, making the capacity of MnO@CNTs much higher than MWCNTs. The energy-power characteristics of the MnO@CNTs film as an electrode for potassium ion half cells are superior to most of the reported

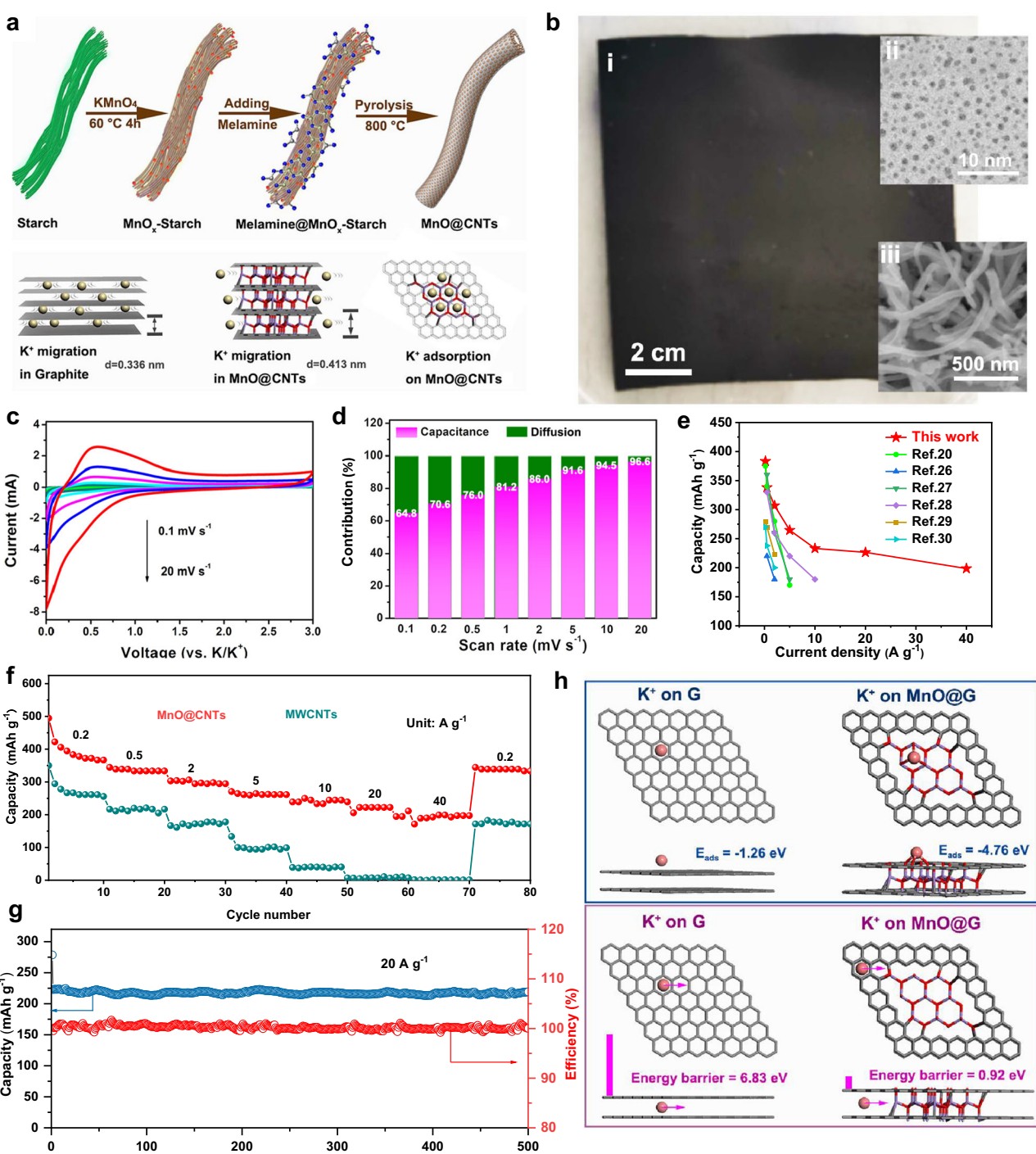

**Fig. 3 | Morphological and electrochemical performance of MnO quantum dots well-inlaid carbon nanotubes (MnO@CNTs) anode. a** The schematic illustration of MnO@CNTs synthesis procedure and the mechanism for K+ storage. **b** The optical images of MnO@CNTs film (**i**), the Cs-corrected STEM image of the MnO QDs (**ii**) and the SEM image of. MnO@CNTs film (**iii**). **c** CV curves of MnO@CNTs electrode at various scan rates of 0.1 to 20 mV s⁻¹. **d** Contribution ratios of the capacitance (pink) and diffusion (green) in the MnO@CNTs-rGO electrode at different scan rates. **e** Comparison of the rate performances of MnO@CNTs-rGO with those of carbon-based anode material for PIBs recently reported. **f** Rate capacities of MnO@CNTs (red) and MWCNTs (green) half-cell at current densities of 0.2, 0.5, 2, 5, 10, 20, and 40 A g⁻¹. **g** Long cycle performance of MnO@CNTs at 20 A g⁻¹. **h** Adsorption energies and energy barriers for K+ ion on graphite and MnO@graphite based on DFT calculations.

materials (Fig. 3e)$^{15,23,30–33}$. Moreover, the MnO@CNTs half-PIHC exhibits excellent cycling with more than 96% capacity retention after 500 cycles at a specific current of 20 A g⁻¹ (~90.66 C according to the capacity, which is equivalent to completing a charge or discharge in 39.6 s) (Fig. 3g). The excellent rate performance of MnO@CNTs is ascribed to the quantum-dot-inlay design and layer-spacing-control,

providing a high pseudocapacitive capacity, which is consistent with the previous literature[34].

The adsorption and migration behavior of K+ ions in MnO@CNTs composites were further explained by DFT theoretical calculations. Figure 3h illustrates the adsorption and migration models of K+ ions on graphene (G) and MnO@CNTs composites. The calculation results

show that the adsorption energy of pure graphene on K+ ions (−1.26 eV) is much higher than that of MnO@CNTs (−4.76 eV), which indicates that MnO@CNTs composites have better compatibility with K+ ions and can effectively promote their adsorption on electrode materials. Moreover, the climbing elastic belt method (CI-NEB) was used to simulate the migration path of K+ ions during the migration process and calculate the transition state and migration energy barrier[34]. The results suggest that the K+ ion in graphene has a higher migration energy barrier (6.83 eV) than that in the MnO@CNTs (0.92 eV) since the interlayer spacing of the graphene layer (-0.34 nm) is smaller than that of the optimized MnO@CNTs composite (-0.41 nm), which also explains that the K+ ions migrate rapidly in the MnO@CNTs composite material and display an excellent rate performance.

## Investigation of the commercial potential of N-CNTs@MC// MnO@CNTs full cell

Before fabricating the ampere-hour-scale pouch cells, the N-CNTs@MC//MnO@CNTs potassium-ion-based full coin cells were assembled to investigate their electrochemical performance and optimize the key parameters (including the mass load of active materials and the amount of electrolyte). The CV curves of optimized N-CNTs@MC//MnO@CNTs full cell at scan rates of 10, 20, 50, 100, and 200 mV s$^{-1}$ are presented in Fig. 4a, which demonstrate that the stable and wide voltage window of N-CNTs@MC//MnO@CNTs full cell ranges from 0 to 4.8 V, ensuring a high energy density. The charge-discharge curves of N-CNTs@MC//K half-cell, MnO@CNTs//K half-cell

and N-CNTs@MC//MnO@CNTs full cell are illustrated in Fig. 4b. The full cell displayed a curve of a slightly deformed isosceles triangle for the charging and discharging processes within a voltage range of 0−4.8 V, indicating an energy storage characteristic of a hybrid supercapacitor. Meanwhile, the charge-discharge curves of N-CNTs@MC//MnO@CNTs full cell at specific currents of 1, 2, 5, 10, 20, and 50 A g$^{-1}$ are shown in Fig. 4c, which all remain the shape of deformed isosceles triangles, suggesting the good reversibility of the full cell at high current density and in a wide voltage window. Moreover, the rate performance of N-CNTs@MC//MnO@CNTs full cell is presented in Fig. 4d, showing that the PIHC delivered the capacity of -120.6, 101.7, 92.3, 78.4, 69.1, and 52.8 mAh g$^{-1}$ at specific currents of 1, 2, 4, 6, 8 and 10 A g$^{-1}$, respectively (based on the mass of the positive electrode). Moreover, the PIHC maintains nearly 90% of its capacity after 10000 cycles (Fig. 4f) at a high specific current of 10 A g$^{-1}$ (-83.33 C), demonstrating remarkable stability and rate performance. To test the fast charging property, the N-CNTs@MC//MnO@CNTs PIHC full cell was charged at a specific current of 10 A g$^{-1}$ and discharged at 0.2 A g$^{-1}$ (Fig. 4e), which demonstrated that only 16 s was required to make the PIHC fully charged while the discharging process could last 800 s (CE is -100%), indicating the N-CNTs@MC// MnO@CNTs full coin cell could be charged at a high rate (within several seconds) and a high specific power of -26 kW kg$^{-1}$ at 10 A g$^{-1}$ (based on the positive electrode) was achieved.

Moreover, it is worth noting that although many supercapacitors could also achieve high power density with fast charging capability, their applications were limited by the obvious

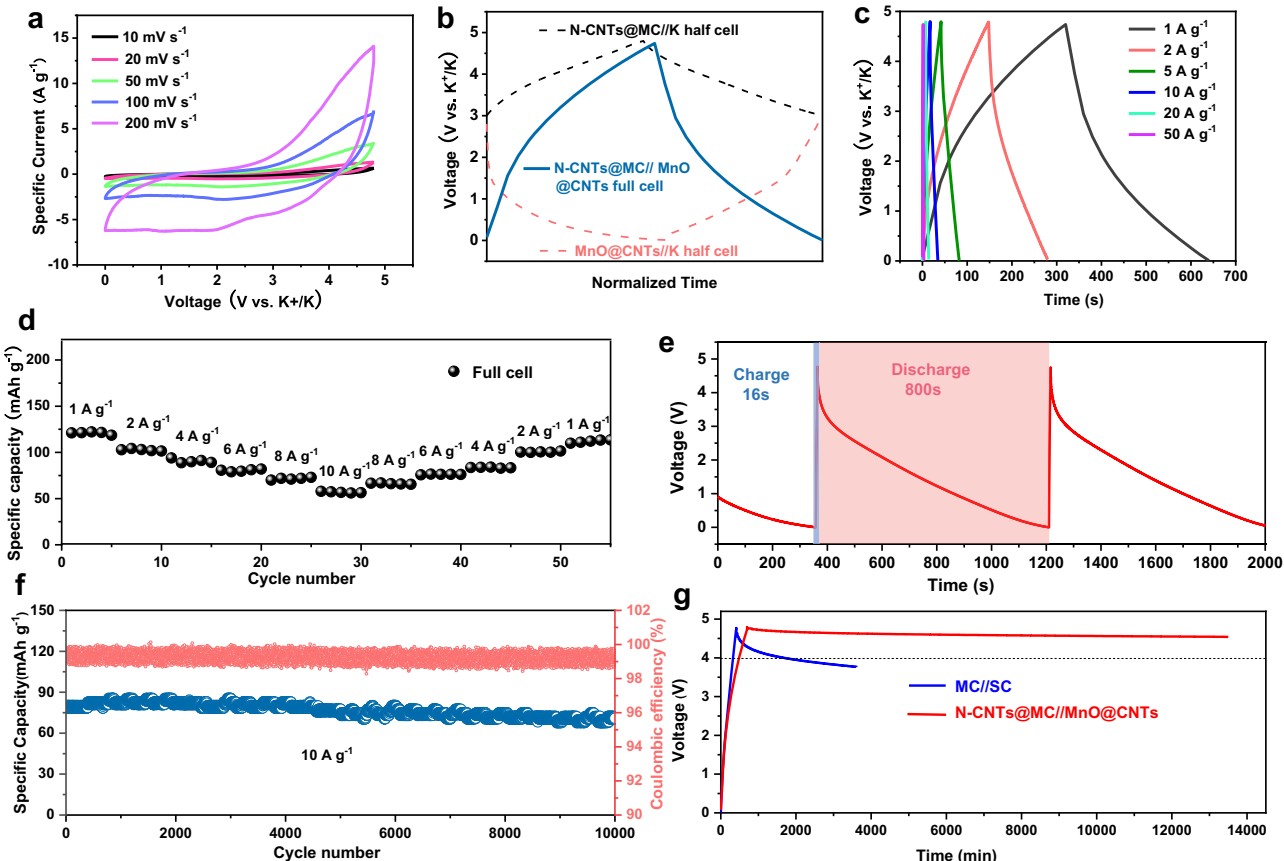

**Fig. 4 | Electrochemical performance of the N-CNTs@MC//MnO@CNTs full coin cell. a** CV curves of N-CNTs@MC//MnO@CNTs full cell at 10, 20, 50, 100 and 200 mV s$^{-1}$. **b** Charge/discharge curves of N-CNTs@MC//K, MnO@CNTs//K half cells and N-CNTs@MC//MnO@CNTs full cell. **c** Charge-discharge curves of N-CNTs@MC//MnO@CNTs full cell at different specific currents. **d** Rate performance of N-CNTs@MC//MnO@CNTs full cell at various specific currents. **e** Fast

charge (blue) and discharge (red) curves of N-CNTs@MC//MnO@CNTs full cell. **f** The long-term cycle performance of N-CNTs@MC//MnO@CNTs full cell at 10 A g$^{-1}$. **g** Self-discharging performance of the N-CNTs@MC//MnO@CNTs full cell in comparison to the mesoporous carbon//soft carbon (MC//SC) potassium-ion-based full.

self-discharge for long-term energy storage[35–37]. In this study, the voltage contention property of the device was evaluated by applying a constant charge to the cell, bringing it to a level of 4.8 V. Subsequently, the voltage of the cell was monitored over an extended period without any further voltage adjustments. Remarkably, the N-CNTs@MC//MnO@CNTs PIHCs exhibited promising voltage retention with an impressively low self-discharge rate (Fig. 4g), which is much better than the meso-porous carbon (MC)//soft carbon (SC) potassium-ion-based full cell[38]. The excellent voltage retention performance should be ascribed to these aspects: *a*. the tunnel-structure of carbon nanotubes in N-CNTs@MC prevents the fast escape of ions; *b*. the N-doping in N-CNTs@MC could fix the stored ions effectively; *c*. the strong adsorption ability of MnO@CNTs on $K^+$ ions is conducive to the low self-discharge rate. X-ray Photoelectron Spectroscopy (XPS) analysis was conducted on the positive electrode before and after 20 cycles from 0 to 4.8 V (Supplementary Fig. 16). The presence of peaks corresponding to C-O and C = O bonds was observed in the positive electrode sample before cycling. However, after 20 cycles, no additional peaks were detected in the positive electrode, indicating the absence of

decomposition of the Dimethyl Carbonate (DMC) or Ethyl Methyl Carbonate (EMC) components. Furthermore, in the F 1*s* curve, the presence of $PF_6^-$ was observed, and no new peaks appeared after 20 cycles, indicating the $PF_6^-$ is stable in this voltage window.

Further, the non-current-collector N-CNTs@MC and MnO@CNTs sheets, which showed excellent electronic conductivity (Supplementary Fig. 18), were used to assemble ampere-hour-level pouch cells to investigate their potential in commercial applications. The approaches to assembling the winding soft pack cells are demonstrated in Fig. 5a–c: the non-current-collector positive electrode and the negative electrode are connected to an Al and Ni tag with a separator interlayer, and then they were wrapped layer by layer in a certain size to reach an ampere-hour capacity, after which these components will be sealed by the aluminum plastic film with a suitable amount of electrolyte in the cell. Due to the optimization of the structure of pouch cells (non-current-collector electrodes) and the electrolyte intake, the overall mass of the 1 Ah pouch cell was decreased to $18.76 \pm 0.5$ g (Supplementary Fig. 19a), achieving high specific energy of about 140 Wh kg$^{-1}$ (based on the whole cell) (Supplementary Fig. 19b) under 10 C rate (where 1 C represents 1 h of testing to charge or discharge fully), which is the highest compared with previously

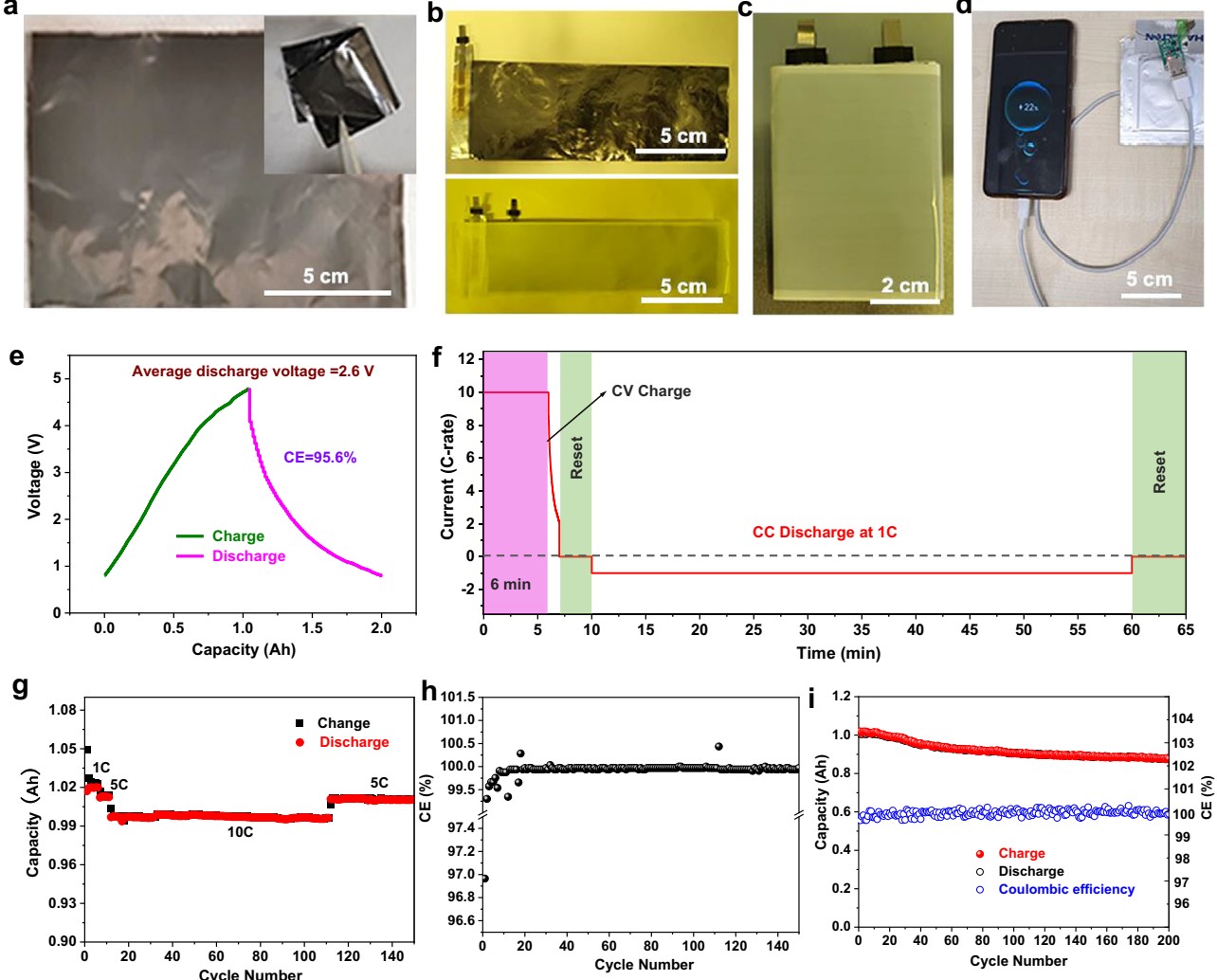

**Fig. 5 | Electrochemical performance of the N-CNTs@MC//MnO@CNTs Ah pouch cell. a** The photograph of a N-CNTs@MC sheet and the exhibition of its flexibility (inset). **b** The optical images of the N-CNTs@MC sheets interconnected with an Al tag and MC//MnO@CNTs sheets with a Ni tag and the separator interlayer. **c** The photograph of a winding soft pack cell before electrolyte injecting and sealing. **d** The photograph of the pouch cell powering a cell phone. **e** Charge-discharge curve of the N-CNTs@MC//MnO@CNT pouch cell. **f** Fast charging profiles of the pouch cell **g** Rate performance and **h** corresponding CE of the Ah pouch cell under different rates. **i** Cycle stability of the pouch cell with charging under 10 C and discharging under 1 C for 200 cycles.

reported ampere-hour-level potassium-ion-based energy storage devices. The excellent performance of the pouch cell is shown as follows. Figure 5e shows the charge-discharge curve of the pouch cell, demonstrating a charge-discharge voltage window from 0.8 to 4.8 V, with an average discharge voltage of -2.6 V with a high CE of 99.98% after activation (first 3 cycles). To measure the fast-charging property of the pouch cell, it was charged at a constant current of 10 C for 5 mins followed by a constant voltage at 4.8 V for 1 min and then discharged at 1 C after a rest (Fig. 5f). The fully charged pouch cell can be used as a power supplier to charge a mobile phone (Fig. 5d). After about 100 cycles at 10 C, the capacity of the pouch cell returned to nearly 1 Ah at 5 C without decay in comparison to the former state under 5 C (Fig. 5g), suggesting the high reversibility of the potassium-ion-based pouch cell. More importantly, the pouch cell delivered excellent cycling stability under a high current density of 10 C with a high average CE of 99.96% after the first 3 cycles (activation period) onward to 150 cycles (Fig. 5h). A longer cycling measurement was conducted as shown in Fig. 5i, illustrating that the pouch cell delivered an initial capacity of 1 Ah and retained up to 0.88 Ah after 200 cycles. Therefore, the N-CNTs@MC//MnO@CNTs full cells show high potential for application in fast-charging energy storage systems due to their superb comprehensive performance of high specific energy (140 Wh kg$^{-1}$), excellent rate performance (fully charged in 6 min) and good cycling stability. The comprehensive performances of PIHCs are well-placed among the state-of-the-art potassium-ion-based energy storage devices (see Supplementary Table 3).

In this work, a holistic optimization for potassium-ion-based devices was conducted to achieve fast chargeable 1Ah PIHCs pouch cells with high energy density, which possess superb comprehensive performance of high specific energy of 140 Wh kg$^{-1}$ (based on the whole mass of device), excellent performance (fully charged in 6 min) and good cycling stability (88% capacity retention after 200 cycles under 10 C). This specific energy is close to that of conventional lithium-ion batteries at 1 C, which is a tremendous breakthrough in the extreme fast charging of Ah-scale energy storage devices, showing the high commercial application in power-consuming portable devices and electric vehicles with rational battery configuration. This breakthrough was accomplished based on some advanced fundamentals. The proposed strategy for modulating pseudocapacitance was successfully implemented through a combination of structural design, involving optimization of porosity and enlargement of spacing, as well as the introduction of N-doping and inlaid MnO to facilitate surface electronic redistribution. Additionally, an effective technique was developed for fabricating lightweight and corrosion-resistant freestanding electrodes, enabling the attainment of a high voltage of 4.8 V and an elevated overall energy density. These freestanding electrodes exhibit excellent mechanical properties and possess exceptional electronic conductivity of $(3.17 \pm 0.01) \times 10\,\char`^\,5$ S m$^{-1}$ at $300 \pm 2$ K, which is comparable to conventional metal current collectors. The strategies of optimization for the electrodes in this work, including enhancing and matching the kinetic dynamics of the anode and cathode respectively, and the fabrication of non-current-collector electrodes, offer valuable insights for the development of fast-charging batteries with high energy density and long cycle life in Li/Na batteries and hybrid capacitors. These strategies can be applied to develop high-performance electrodes based on cheaper materials, like porous carbon and expanded graphite to achieve the goals for low-cost/fast-charge EV batteries in future work.

## Methods
### Materials preparation
**The preparation of Fe$_2$O$_3$-N-CNTs@MC.** The Fe$_2$O$_3$-N-CNTs@MC is derived from biomass, corncob, which served as a basic carbon source.

Firstly, the corncob was thoroughly washed and smashed into fine powders. Then the corncob powders were pretreated by concentrated H$_2$SO$_4$ (98 wt.%) to remove H and O elements in advance. The pretreated corncob powders were mixed with melamine as well as 1 wt.% ferrous sulfate (FeSO$_4$), followed by annealing under a nitrogen atmosphere at 800 °C for 2 h. Finally, the Fe$_2$O$_3$-N-CNTs@MC composite was obtained. The Fe$_2$O$_3$@MC was prepared via the same process of Fe$_2$O$_3$-N-CNTs@MC without the addition of melamine and etching. To prepare the MC, the corncob powders are pre-treated by concentrated H$_2$SO$_4$ (98 wt.%) to remove H and O elements in advance. Then, the pre-treated corncob powders are thoroughly washed with deionized water, after which the dried powders are heated to 800 °C with a heating rate of 10 °C/min.

**The preparation of N-CNTs@MC-rGO sheets.** The N-CNTs@MC was prepared by etching the Fe$_2$O$_3$-C NNTs@MC composite with acid and then recovered through ultra-high temperature annealing. Firstly, the Fe$_2$O$_3$-N-CNTs@MC composite was washed with 1 M HCl solution at 60 °C for 4 h. Then the composite was further annealed at ultra-high temperature (3000 °C) under a nitrogen atmosphere to obtain the N-CNTs@MC. The N-CNTs@MC film was fabricated by mixing 10 wt.% graphene oxide (GO, -100 μm) with N-CNTs@MC powder before annealing to obtain a homogeneous N-CNTs@MC-GO suspension. Then, the N-CNTs@MC-GO suspension was dropped on PET film and evaporated at room temperature to obtain the N-CNTs@MC-GO film. The N-CNTs@MC-GO film was annealed at 1000 °C for 2 h and then at 3000 °C for 1 h in a nitrogen atmosphere. Finally, the N-CNTs@MC-rGO film was achieved by a rolling compression process with controlled thickness.

**The preparation of MnO@CNTs-rGO sheets.** The MnO@CNTs were synthesized by a molecular beam template method. Firstly, 1 g starch was distributed into 200 mL ultra-pure water to generate molecular beams, then 0.5 g KMnO$_4$ was added to oxidize these molecular beams, and the tiny reduction products MnO$_x$ adhered to the molecular beams. After thoroughly washing, 5 g melamine was supplemented to generate a fine mixture, and the mixture was finally annealed at 900 °C for 2 h under a nitrogen atmosphere to obtain the MnO@CNTs. The MnO@CNTs film was prepared with a similar method as the N-CNTs@MC-rGO film.

### Characterizations
The morphologies of samples were examined using scanning electron microscopy (SEM) with a Hitachi S-4800 instrument, transmission electron microscopy (TEM) with a JEM-3100F model, and scanning transmission electron microscopy (STEM, JEOL JEM-F200 HR). In particular, AC-STEM-HAADF (high-angle annular dark-field) imaging was performed on a Titan 80-300 scanning/transmission electron microscope equipped with a spherical aberration corrector, operating at 300 kV. The structural properties were analyzed using X-ray diffraction (XRD) with a Rigaku D/max-2200 pc instrument, Raman scattering spectroscopy with an HR800 system, Fourier-transform infrared (FTIR) spectroscopy with a NICOLET iS10 instrument, and X-ray photoelectron spectroscopy (XPS) with an ESCALAB 250 analyzer. Additionally, the specific surface area and porosity information were determined using a Brunauer−Emmett−Teller (BET) analyzer, specifically the ASAP 2460 model by Mac, America. The mobility, resistivity, and electronic conductivity of the flexible films were measured using an Ecopia Hall Effect Tester (HMS-7000) at a temperature of 300 K to provide insights into the charge transport properties and electronic conductivity of the flexible films.

### Battery assembly and electrochemical test
**The assemble and electrochemical test of N-CNTs@MC//K half cell.** The 2032 type N-CNTs@MC//K half cells were assembled in a

glove box under the Ar atmosphere with the oxygen and water content less than 0.1 ppm. The counter electrode was potassium metal, the separator was glass fiber, and the electrolyte was 1 M KPF$_6$ dissolved in the DMC and EMC (DMC/EMC, 3:4, v/v). The areal loading of N-CNTs@MC electrodes was ~2.0 mg cm². The potassium metal electrode was fabricated by compressing a fresh potassium ingot into slices, with a controlled thickness of 0.5 ± 0.05 mm. Subsequently, the slices were further fashioned into disks with a diameter of 6.0 ± 0.2 mm. The electrolyte added for each coin cell is ~30 μL. The electrochemical tests were carried out on a battery testing system (LAND CT-2001A, China) and an electrochemical workstation (CHI-660C).

**The assemble and electrochemical test of MnO@CNTs//K half cell.** The MnO@CNTs//K half cells were assembled as the same procedures in the 2032-type coin cells. The electrolyte for MnO@CNTs//K half cells was 1 M KPF$_6$ dissolved in the dimethyl carbonate (DMC) and diethyl carbonate (DEC) (DMC/DEC, 1:1, v/v). The areal loading of MnO@CNTs electrodes was ~1 mg cm². The electrolyte added for each coin cell is ~15 μL. The electrochemical tests were carried out on a battery testing system (LAND CT-2001A, China) and an electrochemical workstation (CHI-660C). Besides, the prepotassium of the MnO@CNTs negative electrode could also be conducted in the scale-up MnO@CNTs//K electrochemical reaction cell system with CHI-660C workstation.

**The assemble and electrochemical tests of N-CNTs@MC// MnO@CNTs full cell PIHCS.** For the N-CNTs@MC//MnO@CNTs full cell, the as-prepared N-CNTs@MC were pressed into carbon paper to serve as a positive electrode, with an areal loading of ~3.0 mg cm$^{-2}$. The negative electrode was fabricated by mixing the active material (MnO@CNTs) with 10 wt.% graphene and then pressed into carbon paper and pre-potassiated as in the MnO@CNTs//K electrochemical reaction cell system. The areal loading of MnO@CNTs active material in the negative electrode was ~1 mg cm$^{-2}$. The mass ratio of active materials for the positive electrode and the negative electrode is around 3:1. Before full cell assemble, all of the electrode materials were dried at 60 °C in a vacuum for 12 h, and then the prepotassium of the negative electrode was performed in a glove box with potassium ingot as a counter electrode. The prepotassium of the MnO@CNTs anode was carried out within the scaled-up MnO@CNTs//K electrochemical reaction cell system, by connecting the CHI-660C workstation into an Ar filled glove box with the oxygen and water content less than 0.1 ppm. Specifically, the anode electrode was used as the working electrode, and the potassium ingot was used as both counter, and reference electrode in electrolyte. The electrolyte was made by dissolving 1 M KPF$_6$ into the DMC, DEC, and EMC with a volume ratio of DMC/DEC/EMC = 3:3:4. Subsequently, a cyclic voltammetry (CV) scan was conducted at a scan rate of 0.1 mV s-1 for three to five cycles ranging from 3.0 V to 0 V (vs. K + /K). This process was employed to facilitate the prepotassium into the MnO@CNTs electrode until the curves of two consecutive scans aligned, with a capacity loss of less than 1%. In fabricating the PIHCs the pre- potassiated MnO@CNTs paper was used as the negative electrode and N-CNTs@MC carbon paper was adopted as the positive electrode. The galvanostatic charging/discharging process was performed using a battery testing system (LAND CT-2001A, China) within a voltage range of 0.01–4.80 V. CVs were investigated by an electrochemical workstation (CHI-660C) at different scan rates in a window of 0–4.8 V. The test adopted the constant potential EIS test method, and the potential value was set as the open circuit voltage, the sinusoidal voltage amplitude was set as 10 mV, and the scanning frequency was 10 k -10 m Hz.

The specific energy (E) and specific power (P) of PIHCs were calculated according to the following equations:

$$E = I \int V(t)\mathrm{d}t / m \tag{1}$$

$$P = E/t \tag{2}$$

where m is the total mass of electrodes (The mass is determined based on the working electrode in half coin cells, while for the full coin cell, the calculated mass is based on the positive electrode. As for the pouch cell, the mass calculation is based on the entire device.), I is the discharge current, V(t) is the working voltage, t is the discharge time at the end of discharge after the I$_R$ drop.

The mass is determined based on the working electrode, while for the full coin cell, the calculated mass is based on the positive electrode. As for the 1 Ah pouch cell, the mass of the battery core (including cathode, anode, and separator) is 11.0 ± 0.2 g, the mass of electrolyte is 4.5 ± 0.3 g, and the mass calculation for specific energy is based on the entire device. All the electrochemical measurements were conducted under 25 ± 1 °C. To ensure the reproducibility of the results, more than 10 cells were tested for each electrochemical performance.

### Theoretical calculations

Computer modeling was used to understand the adsorption behaviors of PF$_6^-$ on N-CNTs@MC and the effects of MnO quantum dots on the structure and properties of MnOQD@CHNTs. The Vienna Ab initio Simulation Package (VASP) was used for DFT calculation, with supplied projector augmented wave potentials for core electrons[39–41]. The generalized gradient approximation of Perdew-Becke-Ern-zerhof was used for the exchange-correlation functional[40]. The conjugate gradient algorithm was used in the structural optimization of CNTs, PF$_6^-$ on CNTs, MnO, graphene, graphene@K$^+$, and MnO@K$^+$, providing a convergence of $10^{-5}$ eV in total energy. The atomic structures were fully relaxed in all calculations in graphene, while in MnO the lower half of the atoms were fixed and other atoms were fully relaxed. A vacuum distance larger than 10 Å was used to remove the interaction between successive slabs. The cut-off energy was set to 500 eV with a 5 × 5 × 1 K-point mesh to represent the Brillouin zone. The adsorption energy of the potassium ions was evaluated from the following equations:

$$\Delta E_{ads}(G - K^+) = E_{G@K+} - (E_G + E_{K+}) \tag{3}$$

$$\Delta E_{ads}(MnO@G - K^+) = E_{MnO@G@K+} - (E_{MnO@G} + E_{K+}) \tag{4}$$

where $E_G$@K$^+$ and $E_{MnO@G-K+}$ are the total energies of graphene and MnO after K$^+$ adsorption, $E_G$, $E_{MnO@G}$ and $E_{K+}$ are the energies of graphene, MnO and K$^+$, respectively. The transition state and potassium diffusion pathway calculations were performed using the climbing image nudged elastic band (CI-NEB) method.

Meanwhile, the adsorption behavior of the potassium ion on CNTs was evaluated from the following equations:

$$\Delta E_{ads}(CNTs - PF_{6-}) = E_{CNTs@PF6-} - (E_{CNTs} + E_{PF6-}) \tag{5}$$

$$\Delta E_{ads}(NCNTs - PF_{6-}) = E_{NCNTs@PF6-} - (E_{NCNTs} + E_{PF6-}) \tag{6}$$

### Data availability

The authors declare that all the relevant data are available within the paper and its Supplementary Information file. Additional supporting

data of this study are available from the corresponding author on request. Source data are provided with this paper.

## Code availability

The code for dynamic regulating is available from the corresponding author upon request.

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

## Acknowledgements

This work was financially supported by the National Natural Science Foundation of China (Grant nos. 52274298 (H. Zhou), 51974114 (H. Zhou), 51672075 (Y. Kuang) and 21908049 (Y. Kuang)), and Faraday Institute - Battery Study and Seed Research Project (Rational design and manufacture of stacked Li-CO$_2$ pouch cells) (FIRG052, Y. Zhao). National Supercomputing Center in CHANG-SHA is acknowledged for allowing the use of computational resources including TIANHE-1.

## Author contributions

Conceptualization, H.L. and Y.G.; Methodology, H.L. and Y.G.; Investigation, H.L., Y.G., J.L., H.Z., Y.Z., S.J. and Z.H.; Theoretical calculation, H.L., Writing-original draft, H.L. and Y.G.; Writing- review & editing, H.L., Y.G., J.L., S.J, H.Z., Y.Z. and Y.K.; Funding acquisition, H.L. H.Z. S.L.; Supervision, H.Z. Y.Z. S.J. S.L.

## Competing interests

The authors declare no competing interests.
