## [Peer Review File · Nature Communications]

Ampere-hour-scale soft-package potassium-ion hybrid capacitors enabling 6-minute fast-chargingREVIEWER COMMENTS

Reviewer #1 (Remarks to the Author):

This manuscript represents a quality contribution in the field of electrochemical energy storage systems. Here are some areas for improvement to be able to lead to a publication:

1. Bibliographic references must be completed. In particular, the article on the first prototype of hybrid potassium-ion capacitor isn't cited while cells in pouch cell format were made as in this manuscript. (Annaïg Le Comte and al. – Journal of Power Sources)
2. It is necessary to review the references to figures, which are not always correct.
3. It is mentioned several times to refer to "Experimental procedures", but this section does not exist. This will have to be changed
4. For the self-discharge test, how was the cell charged? A constant current charge only or a constant current charge followed by a constant voltage step?
5. Page 17, the energy density seems calculated according to the total mass of the cell while page 21 the calculation is made from the mass of the two electrodes. This inconsistency needs to be clarified.
6. In the entire manuscript, the longest cycling test presented is 1000 cycles. However, the cyclability of a conventional supercapacitor is one million cycles and that of a hybrid lithium supercapacitor is of the order of 200,000 cycles. Have long-term cycles been carried out with the system presented?
7. The presented results raise the question whether the developed system is a potassium hybrid supercapacitor or a potassium power battery. The authors should argue this.

Reviewer #2 (Remarks to the Author):

This article described the fabrication of Ah-scale soft-package potassium-ion hybrid capacitors with high charging features. The authors also claimed a high full-cell level energy density of 140 Wh/kg, but it's unclear how this number was calculated because the energy density in Table S1 is just 131.9 Wh/kg. Contents in Table S1 were not discussed in the manuscript.

Overall, this article represents an interesting engineering work to develop full devices,

particularly for potassium-ion hybrid capacitors, a relatively new energy system. I don't see clear advancement in fundamentals, for example, new knowledge, concepts, techniques, or even materials.

This manuscript is more suitable for journals with a specific focus on engineering aspects. I would suggest the authors submit this manuscript to another journal.

There are also some specific comments:

1. The cathode material is an ordered nitrogen-doped carbon nanotube array on mesoporous carbon (N-CNTs@MC).

Firstly, there is no indication or characterization for this "ordered" structure.

Secondly, where is the mesoporous carbon? How can we observe the structure illustrated in Fig. 1a and Fig. 2a? how was the pure "MC" prepared?

Thirdly, for the porous structure analysis, the formation of 5 and 30 nm mesopores is not explained. Judging from the nitrogen sorption analysis, this N-CNTs@MC contains large portions of micropore surface area. All this information must be carefully analyzed to understand the reason for good performance rather than generally attributing everything to high porosity.

2. The anode material is MnO@CNTs composite. The MnO dots are of nm size. It's highly impossible to form the structure shown in Fig. 1a, Fig. 3a and Fig. S6 a. The 0.41 nm interlayer space cannot accommodate those MnO dots. The observed 0.41 nm interlayer space is in the short range, possibly due to the low graphitisation. Raman spectra of this material show a strong D band. The XRD pattern didn't show apparent peaks around 24-26 degrees. If the 0.41 nm interlayer space is general in this material, the XRD pattern should reflect this information.

3. The XRD of MnO@CNTs needs more analysis. The average particle size of MnO needs to be calculated. Some peaks are not identified. Are there impurities?

4. The inadequate or incorrect structural characterizations and interpretations for both anode and cathode materials would require a re-visit of the charge storage mechanisms for both anode and cathode, for example, the real role of MnO dots.

Reviewer #3 (Remarks to the Author):

Comments on

Ampere-hour-scale soft-package potassium-ion hybrid capacitors enabling 6-minute fast-charging

This work studies the extreme fast-charging feature of composite electrodes in K-ion hybrid capacitor. Though the outcomes are well known facts such as Nanotube architecture and pore volumes impact on capacitance, but the approach is new and different which is appreciable. However many issues requires clarification and missing information can be included. Since this manuscript is using innovative free-standing electrodes, I recommend the acceptance of this manuscript only upon revision. More concerns should be addressed.

1. Synthesis procedure: in synthesis procedure author have developed material at 3000deg which is quite unusual but good. However at such high temp it is common to use Ar atm rather than N2 atm. Why?
2. In CV: beyond 4.2 V especially during anodic sweep, a sudden increase in current is more prominent, which might be indicating electrolyte oxidation . please explain. Additionally does author has any other data of electrolyte decomposition didn't happen at that higher voltages , such as XPS of used electrode etc.
3. Please explain the method of calculating capacitive vs diffusive current elaborately in supplementary . also please indicate coefficients value.
4. Incase of anode, use of MnO QDs are not explained adequately. In discussion, the authors described the entire phenomenon mainly happening due to capacitive type process for which defects and surface helps. In that case how QDs are helping and conductivity of QDs are questionable.
5. Raman data has been included which is appreciated but the explanation regarding same is

inadequate. Please explain about the composite with Raman spectra data.

6. When using the GO the wt% or V% can be inserted in main result and discussion once for better readability

7. Authors are kindly requested to check for language once again as we could find various grammatic errors and spelling mistakes such as “Founding for funding” and “faction for fraction” etc.

Overall the approach reported here is novel enough and results stated are satisfactory with some logical gap. The article may be considered for publication only after major revision filling those logical gaps.

Response to Reviewers

Response to Reviewer#1

Comment–0: This manuscript represents a quality contribution in the field of electrochemical energy storage systems. Here are some areas for improvement to be able to lead to a publication:

Response to Comment–0:

We express our gratitude to the reviewer for their thoughtful and encouraging feedback on our manuscript. We have incorporated the revisions below in response to the reviewer's enumerated comments and believe that the additional information we have supplied significantly bolsters the quality of our manuscript.

Comment–1: Bibliographic references must be completed. In particular, the article on the first prototype of hybrid potassium-ion capacitor isn't cited while cells in pouch cell format were made as in this manuscript. (Annaïg Le Comte and al. – Journal of Power Sources)

Response to Comment–1:

We thank the reviewer for this considerate comment. We have added some references cited in suitable locations. We have carefully considered your comment and incorporated additional references into appropriate sections of the paper.

The new references are cited in the main texts as follows:

“To address these challenges, potassium-ion hybrid capacitors (PIHCs) were proposed recently, which combine the merits of the high-energy density of KIBs-type anode and the high-power density of capacitor-type cathode^{10, 11, 12.}”

“Pouch cell-type PIHCs have received increased attention lately, the formation protocol has been optimized for further enhancement of their performance in addition to the improved energy density and cycling stability^{16,17.}”

12. *Le Comte A, Reynier Y, Vincens C, Leys C, Azaïs P. First prototypes of hybrid potassium-ion capacitor (KIC): An innovative, cost-effective energy storage technology for transportation applications. Journal of Power Sources 363, 34-43 (2017).*
16. *Qian Y, Li Y, Pan Z, Tian J, Lin N, Qian Y. Hydrothermal “Disproportionation” of Biomass into Oriented Carbon Microsphere Anode and 3D Porous Carbon Cathode for Potassium Ion Hybrid Capacitor. Advanced Functional Materials 31, 2103115 (2021).*

17. *Yvenat M-E, Chavillon B, Mayousse E, Perdu F, Azais P. Development of an Adequate Formation Protocol for a Non-Aqueous Potassium-Ion Hybrid Supercapacitor (KIC) through the Study of the Cell Swelling Phenomenon. Batteries 8, 135 (2022).*

Comment–2: It is necessary to review the references to figures, which are not always correct.

Response to Comment–2:

We thank the reviewer for this comment and apologize for this mistake. We have double-checked all the contents and revised the incorrect numbers which are updated in the main text.

Comment–3: It is mentioned several times to refer to "Experimental procedures", but this section does not exist. This will have to be changed.

Response to Comment–3:

We thank the reviewer for this comment and apologize for this mistake. We have removed the **Experimental procedure**'s part from the **SI** to the main text (**Methods** on page 19). The referred "**Experimental procedures**" on pages 6 and 10 have been revised to "**Methods**" and are highlighted in the main text:

*"To obtain the N-CNTs@MC cathode, the precursor Fe₂O₃-(N-CNTs)@MC was first synthesized by an iron-catalyzed biomass carbon precursor growth approach, after which the Fe₂O₃ particles would be etched to form hollow CNTs (Fig. 2a) (see **Methods**)."*

Comment–4: For the self-discharge test, how was the cell charged? A constant current charge only or a constant current charge followed by a constant voltage step?

Response to Comment–4:

We thank the reviewer for raising this question. In the self-discharge test, we want to know the voltage retention ability. Therefore, only a constant current was used to charge the cell. Specifically, a constant charge was applied to charge the cell to 4.8V after which the voltage of the cell was recorded for a relatively long time without a constant voltage step. This approach allowed us to assess the voltage retention ability of the cell under self-discharge conditions. We have added the following description on page 17 in the main text:

"In this study, the voltage contention property of the device was evaluated by applying a constant charge to the cell, bringing it to a level of 4.8V. Subsequently, the voltage of the cell was monitored over an extended period without any further voltage adjustments."

Comment–5: Page 17, the energy density seems calculated according to the total mass of the cell while page 21 the calculation is made from the mass of the two electrodes. This inconsistency needs to be clarified.

Response to Comment–5:

We thank the reviewer for meticulous reviewing and raising this problem in this manuscript. In terms of the coin cell, the energy density and specific capacity are calculated based on the cathode electrode. When it comes to pouch cells, the mass considered is that of the entire cell. This can provide more valuable information for evaluating the commercial application potential of the cells.

We have revised the description on page 21 to provide greater clarity:

“where m is the total mass of electrodes (The mass is determined based on the working electrode in half coin cells; while for the full coin cell, the calculated mass is based on the cathode electrode. As for the pouch cell, the mass calculation is based on the entire device.)”

Comment–6: In the entire manuscript, the longest cycling test presented is 1000 cycles. However, the cyclability of a conventional supercapacitor is one million cycles and that of a hybrid lithium supercapacitor is of the order of 200,000 cycles. Have long-term cycles been carried out with the system presented?

Response to Comment–6:

We thank the reviewer for meticulous reviewing and providing this comment. We have incorporated the long-term cycling (10000 cycles) in **Fig. R1**. The capacity of the PIHC full cell reaches 79.23 mAh g⁻¹ for the first cycle and maintains nearly 90% of its capacity after 10000 cycles (**Fig. R1**) at a high current density of 10 A g⁻¹ (~83.33C), showing excellent stability and rate performance. Notably, to achieve high energy density, we carefully matched the capacities of both the anode and cathode, which means the anode mass is not excessive. Therefore, the stability of the cell shown in **Fig. R1** is extremely competitive, especially in achieving outstanding power density and high energy density simultaneously. This overwhelming performance highlights its application potential. Given the observed consistent capacity maintenance in the final thousands of cycles, we conducted the tests up to 10000 cycles within the limitations of available revision time.

Fig. R1. The long-term cycle performance of NCNTs@MC//MnO@CNTs full cell at 10 A g^{-1} . (Updated Fig. 4f in the Manuscript)

We also updated the description in the main text as shown below:

“Moreover, the rate performance of N-CNTs@MC//MnO@CNTs full cell is presented in Fig. 4d, showing that the PIHC delivered the capacity of ~ 120.6 , 101.7 , 92.3 , 78.4 , 69.1 and 52.8 mAh g^{-1} at specific currents of 1 , 2 , 4 , 6 , 8 and 10 A g^{-1} , respectively (based on the mass of the cathode electrode). Moreover, the PIHC maintains nearly 90% of its capacity after 10000 cycles (Fig. 4f) at a high specific current of 10 A g^{-1} ($\sim 83.33\text{C}$), demonstrating remarkable stability and rate performance.”

Comment–7: The presented results raise the question whether the developed system is a potassium hybrid supercapacitor or a potassium power battery. The authors should argue this.

Response to Comment–7:

We thank the reviewer for this comment. The distinction between a potassium hybrid supercapacitor and a potassium power battery can be clarified by considering the specific characteristics and design principles of each system. [Chem. Soc. Rev., 2015,44, 1777-1790] A supercapacitor is an energy storage device that stores energy through the separation of charge at the electrode-electrolyte interface. It typically utilizes high surface area electrodes and an electrolyte that allows for rapid ion adsorption and desorption. Supercapacitors are known for their high power density, fast charging/discharging rates, and long cycle life. However, their energy density is relatively lower compared to batteries. On the other hand, a power battery, such as a potassium-ion battery, relies on intercalation and de-intercalation of ions within the solid electrode material during the charging and discharging processes. Batteries generally have higher energy density than supercapacitors but may have lower power density and slower charge/discharge rates. When it comes to the question of whether the developed system is a

potassium hybrid supercapacitor or a potassium power battery, the arguments should be based on the system's design, operating principles, and performance characteristics.

If the system employs high surface area electrodes, such as carbon-based materials, and the emphasis is placed on high power density, fast charge/discharge rates, and long cycle life, it would be reasonable to categorize it as a potassium hybrid supercapacitor. [*Journal of Power Sources*, 2017, 363: 34-43] On the other hand, if the system utilizes solid-state electrodes capable of intercalating and de-intercalating potassium ions, and its design aims for higher energy density while sacrificing some power density and charging/discharging speed, it could be classified as a potassium power battery. [*Energy Storage Materials*, 2021, 41: 108-132.]

In our work, we utilized high surface area carbon-based materials (N-CNTs@MC, with a surface area of $1806.3 \text{ cm}^2 \text{ g}^{-1}$) as electrodes, with a focus on achieving high power density (1400 W kg^{-1}), fast charge/discharge rates (10C), and long cycle life (10,000 cycles). To enhance potassium ion migration, we incorporated MnO quantum dots into multiple-wall carbon nanotubes (MnO@CNTs), which expanded the inter-layer of CNTs. As a result, we observed that the capacity of MnO@CNTs was primarily governed by the capacitive contribution, rather than intercalation and de-intercalation processes, especially under high-rate conditions (capacitive contribution reaching up to 96.6% at 20 A g^{-1}).

In conclusion, based on the provided explanation of the system's design, the utilization of high surface area carbon-based materials (N-CNTs@MC), the emphasis on high power density, fast charge/discharge rates, and long cycle life, as well as the evidence from experimental data and electrochemical performance, it is evident that the developed system can be classified as a **potassium hybrid supercapacitor**. The use of carbon-based materials with high surface area suggests a capacitive energy storage mechanism, while the observed high power density, fast charge/discharge rates, and long cycle life further support its supercapacitor characteristics. Therefore, the overall evidence presented in our work strongly indicates the **potassium hybrid supercapacitor** nature of the developed system.

We have added the following description on page 5 and cited references 12 and 13 to provide greater clarity:

“Although a holistic optimization was achieved in the system, enabling the device to deliver specific energy comparable to the K ion battery, the system is classified as a potassium hybrid supercapacitor due to their energy storage mechanism.^{12, 13}”

12. Le Comte A, Reynier Y, Vincens C, Leys C, Azaïs P. First prototypes of hybrid potassium-ion capacitor (KIC): An innovative, cost-effective energy storage

technology for transportation applications. Journal of Power Sources **363**, 34-43 (2017).

13. Chang C-H, Chen K-T, Hsieh Y-Y, Chang C-B, Tuan H-Y. Crystal facet and architecture engineering of metal oxide nanonetwork anodes for high-performance potassium ion batteries and hybrid capacitors. *ACS nano* **16**, 1486-1501 (2022).

Response to Reviewer#2

Comment–0: This article described the fabrication of Ah-scale soft-package potassium-ion hybrid capacitors with high charging features. The authors also claimed a high full-cell level energy density of 140 Wh/kg, but it's unclear how this number was calculated because the energy density in Table S1 is just 131.9 Wh/kg. Contents in Table S1 were not discussed in the manuscript.

Overall, this article represents an interesting engineering work to develop full devices, particularly for potassium-ion hybrid capacitors, a relatively new energy system. I don't see clear advancement in fundamentals, for example, new knowledge, concepts, techniques, or even materials.

This manuscript is more suitable for journals with a specific focus on engineering aspects. I would suggest the authors submit this manuscript to another journal.

Response to Comments–0:

We thank the reviewer for the thoughtful comments on our manuscript. We have addressed the reviewer's comments as shown below and believe that our revisions improve the clarity and quality of our manuscript.

Firstly, we apologize for the mismatched information in **Supplementary Table 1**. The energy density of the pouch cell is indeed 140 Wh/kg, as previously reported in the manuscript. We discovered that the initial calculation result of 131.9 Wh/kg was calculated in a different method. After we unified the calculation method, we made the necessary correction to the energy density value in the main text, but we inadvertently failed to update **Supplementary Table 1** in the Supporting Information section. We now have replaced the old **Supplementary Table 1**. The parameters in the new table are the same as the old one, except for the number of the energy density. The equation of the calculation method of the energy density is described in the **Methods** section of the manuscript. $E = I \int V(t) dt/m = C \bar{U}/m = (1.01 \text{ Ah} \times 2.6 \text{ V}) / (18.7584 \times 10^{-3} \text{ Kg}) \approx 140 \text{ Wh/kg}$, where C is the capacity of the cell, U is the average voltage and m is the total mass of the full cell.

Updated Supplementary Table 1. Optimized parameters of the soft-packaged PIHCs with the capacity of ~1Ah

Parameters of wound soft-package PIHCs						
	Cathode	Anode	Separator	S (cm ²)	Total V (cm ³)	Volume Density (Wh/L)
Area mass (mg cm ⁻²)	30.0	10.0	1.7	Single	7.5	283.2
				24.8		
Thickness (μm)	150.0	40.0	12.0	Stack	Electrolyte (mg)	Shell and Tags (mg)
				10.0	4474.5	3186.4
Denisty (g cm ⁻³)	~2.0	~2.5	~1.42	Core mass	Total mass (mg)	Energy Density (Wh/Kg)
Mass (mg)	7425.0	2722.5	950.0	11097.5	18758.4	140.0

Secondly, we proposed a holistic optimization strategy to achieve ampere-hour-scale pouch cells with dramatically enhanced energy density, power density and voltage retention capability. This exciting result was achieved based on some advancement in fundamentals, rather than relying solely on engineering development.

1. To address the mismatched kinetic, which is the serious intrinsic problem of potassium-ion hybrid capacitors (PIHCs), we proposed **a new concept – modulating pseudocapacitance** to balance the capacity and kinetics mismatch of capacitor-type cathode and battery-type anode. To be specific, a strategy was developed to simultaneously modulate surface redox kinetics at the cathode and enhance ion intercalation while introducing pseudocapacitive active sites at the anode, enabling both to exhibit pseudocapacitive behavior – continuous and fast reversible redox reactions or intercalation at/near the surface of electrode materials. To prove the concept, we fabricated an ordered nitrogen-doped carbon nanotube array on mesoporous carbon (N-CNTs@MC), where the surface redox pseudocapacitance was introduced by N-doping and defect engineering. For the anode, MnO quantum dots inlaid carbon nanotubes (MnO@CNTs) were synthesised, where the MnO quantum dots enhance the pseudocapacitance and enlarge the layer spacing which can improve the intercalation kinetics (*the previous schematic illustration is inadequate, which might affect your understanding of this manuscript, we have clarified it in the following comments*). Additionally, the advancement of surface electron redistribution, induced by defects

(resulted from N-doping and inlaid MnO), has been demonstrated through a combination of DFT calculations and experimental validation.

2. While the methods for fabricating these materials may not be new, they represent a new addition to the field of PIHCs and have been optimized to a high degree. To further increase the energy density, **a novel technique** was developed to prepare the freestanding electrodes which are lightweight and corrosion-resistant, enabling a high voltage of 4.8V and higher overall energy density. It is a novel technique rather than a simple mixing method since it addresses the challenge of achieving both high mechanical properties and excellent conductivity. The size distribution of the GO, the gas environment and the high temperature for the annealing process have been investigated thoroughly to optimize the technique.
3. The prepared free-standing electrode is a **new material** in the battery field since the conductivity of these free-standing electrodes reached $3.17 \times 10^5 \text{ S m}^{-1}$ (**Fig. R2**), which is comparable to the conventional metal current collector (**Fig. R3**). To the best of our knowledge, no previous research has successfully achieved a freestanding electrode with conductivity at such an elevated level. This remarkable achievement represents a significant breakthrough as it effectively eliminates the contact resistance typically encountered in conventional coating processes (**Fig. R3b**). The cells prepared by these electrodes deliver excellent rate performance (comparable to supercapacitors) and exhibit high energy density and voltage convention ability (comparable to batteries) (**Fig. R3e, f**). Attaining such an outcome is exceptionally challenging, further highlighting the importance and impact of this result.
4. In addition, the parameters of the electrolyte (including the components and proportions) were systematically optimized to achieve a high voltage of 4.8 V. This achievement is particularly noteworthy considering the prevailing trend in reported Li-ion batteries, which typically operate under 4.5V. This optimization strategy was driven by the organic molecules model rather than an empirical adjustment of the E/C ratio, as commonly used in engineering approaches.

As a result, 1 Ah soft-package PIHCs could exhibit a high voltage of up to 4.8 V and specific energy of 140 W h kg^{-1} based on the whole weight of PIHC pouch cell (close to the commercial LiFeO₄ batteries) under a rapid charging at 10 C (several times higher than commercial Li-ion battery pouch cell). It is a big advancement in PIHCs for practical application, and also a breakthrough in extreme fast charging of Ah-scale energy storage devices. This represents a significant advancement in the practical application of PIHCs. Therefore, we think this

manuscript is suitable for the comprehensive journal of *Nature Communications*.

We have added the following description on the summary part of the main text (page 20) to provide greater clarity:

“This breakthrough was accomplished based on some advanced fundamentals. The proposed strategy for modulating pseudocapacitance was successfully implemented through a combination of structural design, involving optimization of porosity and enlargement of spacing, as well as the introduction of N-doping and inlaid MnO to facilitate surface electronic redistribution. Additionally, a novel technique was developed for fabricating lightweight and corrosion-resistant freestanding electrodes, enabling the attainment of a high voltage of 4.8V and an elevated overall energy density. These freestanding electrodes exhibit excellent mechanical properties and possess exceptional conductivity ($3.17 \times 10^5 \text{ S m}^{-1}$) which is comparable to conventional metal current collectors. The strategies of optimization for the electrodes in this work, including enhancing and matching the kinetic dynamics of the anode and cathode respectively, and the fabrication of non-current-collector electrodes, offer valuable insights for the development of fast-charging batteries with high energy density and long cycle life in Li/Na batteries and hybrid capacitors.”

Fig. R2. The result of the Hall measurement on the N-CNTs@MC film. (Updated Supplementary Fig. 20)

Fig. R3. Comparison of the conductivity of **a** Cu foil, **b** the anode material coated on Cu foil, **c** N-CNTs@MC film and **d** MnO@CNTs film. **e** the photograph of the coin cell lighting a LED light, **f** the self-discharge test of the coin cell. (Updated **Supplementary Fig. 18**)

Comment–1: The cathode material is an ordered nitrogen-doped carbon nanotube array on mesoporous carbon (N-CNTs@MC).

Firstly, there is no indication or characterization for this “ordered” structure. Secondly, where is the mesoporous carbon? How can we observe the structure illustrated in Fig. 1a and Fig. 2a? how was the pure “MC” prepared?

Thirdly, for the porous structure analysis, the formation of 5 and 30 nm mesopores is not explained. Judging from the nitrogen sorption analysis, this N-CNTs@MC contains large portions of micropore surface area. All this information must be carefully analyzed to understand the reason for good performance rather than generally attributing everything to high porosity.

Response to Comment–1:

We thank the reviewer for the thoughtful comments on our manuscript.

First, the structure of ordered nanotubes, which was not previously clear, has been visualized in **Fig.R4**. The “ordered” means that the carbon nanotubes are orientally distributed. This figure and the following description have been added to the **Supporting Information** and main text:

*“After acid-washing and heat treatment at 3000°C under a nitrogen atmosphere, the N-CNTs@MC composite was obtained (**Fig.R4**). The SEM image of the nitrogen-doped carbon*

nanotube reveals the nanotube array exhibiting a discernible level of structural organization. The nanotubes display a degree of orderliness and the individual nanotubes are discernible and appear uniformly distributed.”

Fig.R4. SEM images of N-CNTs@MC after acid wash and high-temperature heat treatment at different magnification, scale bar: **a** 2 μm , **b** 2 μm , **c**. 0.2 μm , **d** 500 nm (pink circle: open end of the CNTs). (Supplementary Fig. 5)

Fig. R5. The schematic illustration of the process of preparing the N-CNTs@MC. (Updated Supplementary Fig. 2)

The pure “MC” is the porous carbon matrix with some mesopores, which is prepared by annealing the pre-treated carbon powder (MC precursor). Specifically, the corncob powders are pre-treated by concentrated H_2SO_4 (98 wt.%) to remove H and O elements in advance. Then, the pre-treated corncob powders are thoroughly washed with deionized water, after which the dried powders are heated to 800 $^\circ\text{C}$ with a heating rate of 10 $^\circ\text{C}/\text{min}$. As a result, the porous carbon matrix can be obtained.

In terms of the N-CNTs@MC, the Fe catalytic sites are dispersed on the **MC precursor**, and then the powders are mixed with melamine. During the annealing process, the Fe atoms serve as the catalytic site for CNTs growth, while melamine serves as the source of nitrogen (N) and carbon (C) for the formation of N-CNTs. As a result, an ordered nitrogen-doped carbon nanotube array on mesoporous carbon (N-CNTs@MC) is obtained. The schematic illustration of the process is shown in **Fig. R5**.

For comparison purposes, Fe₂O₃@MC was prepared without the addition of melamine, and the morphologies of Fe₂O₃@MC and N-CNTs@MC are shown in **Fig. R6c, d**, indicating the Fe are dispersed on the MC substrate and the CNTs grow on the substrate.

Fig. R6. a The SEM of the MC; **b** the Nitrogen adsorption-desorption isotherms and pores size distribution of the MC; **c** The SEM of Fe₂O₃@MC; **d** the SEM of the N-CNTs@MC. (Supplementary Fig. 10)

The relative figures and the following description have been added to the main text and **Supporting Information**:

(Page 6 in the main text): “To obtain the N-CNTs@MC cathode, as illustrated in **Supplementary Fig. 2**, the precursor Fe₂O₃-(N-CNTs)@MC was first synthesized by an iron-catalyzed biomass carbon precursor growth approach, after which the Fe₂O₃ particles would be etched to form hollow CNTs (Fig. 2a) (see **Methods**). For the comparison, the Fe₂O₃@MC was prepared without the addition of melamine and etching, while the MC was prepared by

annealing the pre-treated carbon powder (MC precursor).”

(Page 21 in the Methods): “The Fe₂O₃@MC was prepared via the same process of Fe₂O₃-N-CNTs@MC without the addition of melamine and etching. To prepare the MC, the corncob powders are pre-treated by concentrated H₂SO₄ (98 wt.%) to remove H and O elements in advance. Then, the pre-treated corncob powders are thoroughly washed with deionized water, after which the dried powders are heated to 800 °C with a heating rate of 10 °C/min.”

Thirdly, we conducted another round of the N₂ adsorption-desorption isotherm measurement and further optimized the curve fitting to analyze the distribution of the pores. The updated figure (**Fig. R. 7b**) shows that the N-CNTs@MC sample contains a small fraction of micropores (~2 nm). These micropores and ~5 nm pores come from MC, while the pores with a diameter of 30 nm are attributed to the presence of CNTs. To confirm this, we measure the MC samples, **Fig. R6a** shows the SEM image of the MC sample, and the distribution of the pores is shown in **Fig. R6b**. The MC reveals the presence of micropores (~2 nm) and mesopores (~5 nm). With the growth of N-CNTs on the MC, additional pores (~30 nm) emerge (**Fig. R 7b**), which is attributed to the formation of hollow carbon nanotubes.

Fig.R7 a Nitrogen adsorption-desorption isotherms and **b** Pore size distributions of N-CNTs@MC (insert: The enlarged illustration of the marked area). (**Supplementary Fig. 11**)

The good performance of the N-CNTs@MC is attributed to the optimized porous structure and alternation of electrons distribution via N doping. To understand the contribution of N doping, we conducted DFT calculations to analyze the adsorption behaviour and redox mechanism of PF⁶⁻ ions on CNTs and N-doped CNTs (**Fig. 2j** and **Supplementary Fig. 13**). The results showed that the N-CNTs exhibits higher adsorption of PF⁶⁻, which is beneficial to obtaining fast charge/discharge processes and higher capacitance. To comprehend the porosity, we first

conducted a comparison between the performance of N-MC (synthesized using the same method but without Fe catalysts) and N-CNTs@MC (as shown **Fig. R8a**). This comparison clearly demonstrates that the inclusion of the N-CNTs array significantly enhances the performance of MC. This improvement can be primarily attributed to the increased specific surface and the incorporation of larger nanopores (~30 nm). Additionally, to optimize the porous structure, we examined the performance of N-CNTs@MC fabricated with different ratios of MC precursor and melamine which affect the portions of nanopores (**Fig. R8b**). The result shows that the 1:2 sample exhibits the best performance among these samples.

Fig. R8 a Rate performance of the MC and N-CNTs@MC samples, **b** The comparison of the rate performance of N-CNTs@MC prepared with different ratios of MC precursor and melamine. (**Supplementary Fig. 12**)

We have added the following description in the main text (page 20) to provide greater clarity:

(Page 7): “The BET-specific surface area of N-CNTs@MC was calculated to be 1806.3 m² g⁻¹ and the pore size distribution was concentrated on ~5 nm and ~30 nm as well as some micropores (~2 nm). These micropores and ~5 nm pores come from MC, while the pores with a diameter of 30 nm are attributed to the presence of CNTs.”

(Page 8): “To comprehend the porosity, we first conducted a comparison between the performance of N-MC (synthesized using the same method but without Fe catalysts) and N-CNTs@MC (as shown **Supplementary Fig. 12a**). This comparison clearly demonstrates that the inclusion of the N-CNTs array significantly enhances the performance of MC. This improvement can be primarily attributed to the increased specific surface and the incorporation of larger nanopores (~30 nm). Additionally, to optimize the porous structure, we examined the performance of N-CNTs@MC fabricated with different ratios of MC precursor and melamine which affect the portions of nanopores (**Supplementary Fig. 12b**). The result shows that the 1:2

sample exhibits the best performance among these samples.”

Comment–2: The anode material is MnO@CNTs composite. The MnO dots are of nm size. It's highly impossible to form the structure shown in Fig. 1a, Fig. 3a and Fig. S6 a. The 0.41 nm interlayer space cannot accommodate those MnO dots. The observed 0.41 nm interlayer space is in the short range, possibly due to the low graphitisation. Raman spectra of this material show a strong D band. The XRD pattern didn't show apparent peaks around 24-26 degrees. If the 0.41 nm interlayer space is general in this material, the XRD pattern should reflect this information.

Response to Comments–2:

We thank the reviewer for carefully reviewing and bringing this issue to our attention. Here, we apologize for any confusion caused by the previously provided figures. The purpose of the schematic figures was to visually convey the concept of expanding the interlayer space.

Fig. R9. *The illustration of the comparison between MWCNTs and MnO@CNTs. (Supplementary Fig. 8)*

Indeed, it is true that the interlayer space of 0.41 nm cannot accommodate the entire MnO dots. Individual MnO dots are embedded between multiple carbon layers, and their atomic structure has been validated through DTF modeling. While MnO crystals possess a relatively rigid crystal structure, the spacing between graphite carbon layers is more flexible and adjustable. When MnO is embedded within the carbon layers, a relatively stable heterojunction is formed between the carbon layer and MnO. The minimum layer spacing of MnO is approximately 0.21 nm.

Consequently, each 2-carbon-layer segment aligns well with a 3-atomic-layer segment of MnO, resulting in a spacing of approximately 0.41 nm. This alignment is consistent with the characterization results, as illustrated in **Fig. R9 (Supplementary Figure S8)**.

In the majority of cases, the MnO dots are encapsulated by four carbon layers (equivalent to three interlayers), which aligns with the calculated sizes of the MnO dots (**Fig. R14**). The intercalation of MnO leads to an expansion of the spacing between the carbon layers from 0.34 nm to 0.41 nm. It is noteworthy that the crystal plane of MnO and the graphite's six-membered ring structure exhibit favorable lattice parameter matching, promoting the formation of stable heterostructures, as depicted in **Fig. R9f (Supplementary Figure S8f)**.

Fig. R10. Adsorption energies and energy barriers for K^+ ion on graphite and MnO@graphite based on DFT calculations. (Updated **Fig. 3h**)

The stable structure of MnO@CNTs has been thoroughly modeled and calculated using DTF, as demonstrated in **Fig. R10**. Between two carbon layers, each carbon interlayer can accommodate a three-atomic-layer segment of the MnO crystal, rather than the entire MnO dots. Furthermore, the functions of the embedded MnO quantum dots were revealed through DFT calculations. Specifically, the adsorption energies between K ions and the MnO located within

the carbon layer, as well as the energy barrier for K ion transfer in the interlayer, were calculated. These calculations indicate that the embedding of MnO QDs within the carbon layers leads to an increased adsorption energy towards K ions and a significantly reduced energy barrier for K ion migration. Consequently, the introduction of MnO into the carbon layers not only enhances the capacity but also improves the kinetics and rate performance for the PIHCs. To alleviate any misunderstandings arising from the schematic figures, we have made updates to **Fig. 1a, iii**, **Fig. 3a** and **Supplementary Fig. 7a**. as well.

Fig. R11. Holistic design and optimisation strategy of PIHCs and comparison with commercial energy storage cells. (Updated Fig. 1)

Fig.R12. The schematic illustration of MnO@CNTs synthesis procedure and the mechanism

for K^+ storage. (Updated Fig. 3a)

Fig.R13. *a* Schematic diagram, *b* TEM, and *c* enlarged TEM images of MnO@CNTs; *d* High resolution TEM image on the wall of MnO@CNTs; *e* Atomic image of the MnO quantum dots; *f* TEM image of MnO@CNTs with clear diffraction fringes of carbon layer and *g* corresponding carbon layer spacing. (Supplementary Fig. 7)

In addition, we thank the reviewer for addressing the query for the XRD result. We have conducted a thorough re-evaluation of the data and performed remeasurements on certain samples. Here, we have updated the data which are shown in **Fig.R14 (Supplementary Fig. 14)**. The final MnO@CNTs exhibit prominent MnO peaks. When compared to the CNTs, the MnO@CNTs display a broad peak between 24-26 °C, which is attributable to defects resulting from the embedded MnO in the carbon layer and nitrogen doping. Moreover, the peak of MnO@CNTs exhibits a slight leftward shift compared to the CNTs, indicating an enlarged interlayer spacing, consistent with the structural analysis conducted via TEM.

Below is the updated description in the main text (page 11):

“The XRD curve of MnO@CNTs exhibit prominent MnO peaks. When compared to the CNTs,

the MnO@CNTs display a broad peak between 24-26 °C, which is attributable to defects resulting from the embedded MnO in the carbon layer and nitrogen doping. Moreover, the peak of MnO@CNTs exhibits a slight leftward shift compared to the CNTs, indicating an enlarged interlayer spacing, consistent with the structural analysis conducted via TEM.”

Fig.R14. XRD patterns of MnO@CNTs and CNTs: **a** from 22° to 30°, **b** from 30° to 80°. **c** Raman spectra of MnO@CNTs. (Updated **Supplementary Fig. 14**)

Comment–3: The XRD of MnO@CNTs needs more analysis. The average particle size of MnO needs to be calculated. Some peaks are not identified. Are there impurities?

Response to Comment–4:

We thank the reviewer for careful reviewing and providing valuable comments. We have conducted a thorough re-evaluation of the data and performed remeasurements on certain samples. The XRD figure of MnO@CNTs has been updated in **Fig. R14 (Supplementary Fig. 14)**.

The final MnO@CNTs exhibit prominent MnO peaks. When compared to the CNTs, the MnO@CNTs display a broad peak around 24-26 °C, which is attributable to defects resulting from the embedded MnO in the carbon layer and nitrogen doping. Moreover, the peak of MnO@CNTs exhibits a slight leftward shift compared to the CNTs, indicating an enlarged interlayer spacing, consistent with the structural analysis conducted via TEM. Except for these peaks, there are no other discernible peaks, indicating a high level of purity.

In terms of the particle size of MnO, we have counted an analysis on the size distribution of MnO quantum dots (QDs) in the wall of MnO@CNTs. The resulting curve illustrating this distribution is depicted in **Fig. R15b**. The average diameter of the MnO QDs was calculated to be approximately 1.4 nm, indicating that these QDs are accommodated within a space equivalent to approximately four layers of carbon (corresponding to approximately three interlayers).

Fig. R15. *a* Cs-corrected STEM image of the MnO QDs in the wall of MnO@CNTs. *b* The corresponding size distribution of the MnO QDs.

Comment-4: The inadequate or incorrect structural characterizations and interpretations for both anode and cathode materials would require a re-visit of the charge storage mechanisms for both anode and cathode, for example, the real role of MnO dots.

Response to Comment-4:

We thank the reviewer for carefully reviewing and providing valuable comments. We apologise that the previous schematic diagrams and inappropriate XRD curve might lead to potential misunderstandings on this work. We have revised **Fig. 1a**, **Fig. 3a** and **Supplementary Fig. 7a**, and updated a better XRD figure as shown in **Fig.R14 (Supplementary Fig. 14)**. In terms of the storage mechanisms for both anode and cathode, we conducted a comprehensive analysis. For the cathode, the optimized porosity and structures can enhance the capacity and rate performance, while the surface redox pseudocapacitance introduced by N-doping or defect engineering enable better compatibility with the anode.

To comprehend the porosity, we first conducted a comparison between the performance of N-MC (synthesized using the same method but without Fe catalysts) and N-CNTs@MC as shown in **Fig. R8 (Supplementary Fig. S12a)**. This comparison clearly demonstrates that the inclusion of the N-CNTs array significantly enhances the performance of MC. This improvement can be primarily attributed to the increased specific surface and the incorporation of larger nanopores (~30 nm). This comparison clearly demonstrates that the inclusion of the N-CNTs array significantly enhances the performance of MC. This improvement can be primarily attributed to the increased specific surface and the incorporation of larger nanopores (~30 nm). Additionally, to optimize the porous structure, we examined the performance of N-CNTs@MC fabricated with different ratios of MC precursor and melamine which affect the

portions of nanopores (**Fig. R8b**). The result shows that the 1:2 sample exhibits the best performance among these samples.

As for the contribution of the N doping, the CV curves (**Fig.2d, e**) were conducted to prove the pseudocapacitance behavior. The incorporation of N introduces additional capacitive storage mechanisms, which can further enhance the energy density of the electrode. Additionally, DFT calculations were conducted to further understand the adsorption behaviour and redox mechanism of PF_6^- ions on N-doped CNTs as shown in **Fig. R16 (Supplementary Fig. 13)**, indicating that the N-doped CNTs possessed a higher adsorption ability (-7.21 eV) to PF_6^- than pure CNTs.

Fig. R16. The optimized structures of adsorbed PF_6^- ions on CNT and N-CNT. (**Supplementary Fig. 13**)

In terms of the anode, we developed the MnO quantum dots well-inlaid carbon nanotubes (MnO@CNTs) were developed. The embedded MnO quantum dots can enlarge the interlayer space of carbon layers, which can significantly enhance the intercalation kinetics to improve the rate performance. In addition, the introduction of MnO can provide enhanced pseudocapacitance at the same time to further improve the whole kinetics, enable better compatibility with the cathode. Firstly, the MnO quantum dots and the enlarged spacing can be observed from TEM measurement (**Fig.R13**). Furthermore, the XRD analysis (**Fig.R14**) provides additional evidence supporting these findings. The DFT calculation has been conducted to analyse the adsorption and migration behaviour of K^+ ions in MnO@CNTs (**Fig. R10**). The results showed that MnO@CNTs composites have better compatibility with K ions

and can effectively promote their adsorption on electrode materials, and the K^+ ions can migrate rapidly in $MnO@CNTs$ compared with CNTs. The pseudocapacitance contribution of $MnO@CNTs$ composite have been investigated as well (**Fig. 3c, d**). The enhanced intercalation kinetics and pseudocapacitance can significantly improve the capacity and rate performance of CNTs. As a result, the energy-power characteristics of the $MnO@CNTs$ film as an anode electrode for potassium ion half cells are superior to most of the reported materials (**Fig. 3e**) The combination between the matched cathode and anode can achieve high capacity and high-rate performance simultaneously.

The updated description of the contributions of anode and cathode is as below:

(Page 5):“The optimized porosity and structures can enhance the capacity and rate performance, while the surface redox pseudocapacitance introduced by N-doping or defect engineering enable better compatibility with the anode. For the anode, the MnO quantum dots inlaid carbon nanotubes ($MnO@CNTs$) were synthesised as the electrode. The MnO quantum dots inlaid in the CNT walls can enlarge the interlay space of carbon layers, which can significantly enhance the intercalation kinetics to improve the rate performance^{19, 20}. In addition, the introduction of MnO can provide enhanced pseudocapacitance at the same time to further improve the whole kinetics, enable better compatibility with the cathode. The combination between the matched cathode and anode can achieve high capacity and high-rate performance simultaneously.”

Response to Reviewer#3

Comment–0: This work studies the extreme fast-charging feature of composite electrodes in K-ion hybrid capacitor. Though the outcomes are well known facts such as Nanotube architecture and pore volumes impact on capacitance, but the approach is new and different which is appreciable. However many issues requires clarification and missing information can be included. Since this manuscript is using innovative free-standing electrodes, I recommend the acceptance of this manuscript only upon revision. More concerns should be addressed.

Response to Comment–0:

We express our gratitude to the reviewer for their thoughtful and positive comments on our manuscript. We have incorporated the revisions below in response to the reviewer's enumerated comments and believe that the additional information we have supplied significantly improve the quality of our manuscript.

Comment–1: Synthesis procedure: in synthesis procedure author have developed material at 3000deg which is quite unusual but good. However, at such high temp it is common to use Ar atm rather than N₂ atm. Why?

Response to Comment–1:

We thank the reviewer for careful reviewing and raising this question.

It's true that in most high-temperature synthesis procedures, especially those performed at 3000 degrees, an argon atmosphere is more commonly preferred due to its inertness, thermal stability, higher boiling point, and better thermal conductivity. However, the use of a nitrogen (N₂) atmosphere is more specialized and typically reserved for situations where its specific properties align with the requirements of the synthesis procedure.

In our case, there are certain situations where a nitrogen atmosphere can offer specific benefits:

- 1. Chemical reactions:** Nitrogen can participate in certain chemical reactions that can be advantageous in specific synthesis procedures. In our case, the cathode electrode material N-CNTs@MC is a nitrogen-doped carbon material, and the introduction of nitrogen atoms enhances the pseudo-capacitance, thereby improving the capacity. Since our goal is to maximize the nitrogen doping in our materials, the synthesis procedure specifically requires the presence of nitrogen as an active component. Using a nitrogen atmosphere can facilitate these reactions by providing a suitable environment for nitrogen incorporation into the material structure. Nitrogen can act as a reactant or a

carrier gas in reactions involving nitrogen-based compounds, enabling the desired nitrogen doping in the carbon material. Therefore, in this specific case, utilizing a nitrogen atmosphere in the synthesis procedure is essential to achieve the desired nitrogen doping levels and enhance the pseudo-capacitance of the cathode electrode material. It allows for the optimized incorporation of nitrogen, thereby improving the capacity and performance of the material.

- 2. Reduced contamination:** Nitrogen is often considered a cleaner gas compared to argon. Argon can contain trace amounts of impurities, including oxygen, moisture, or other gases. In certain synthesis procedures, especially those that are highly sensitive to impurities, using a nitrogen atmosphere can be beneficial to minimize contamination and ensure the production of a purer final product. By using a nitrogen atmosphere, we can create a controlled and purified environment for the synthesis procedure. Nitrogen gas helps prevent unwanted reactions or interactions with the materials being synthesized. This reduces the risk of introducing impurities and ensures the integrity of the final product. In our specific case, where the synthesis procedure aims to develop a high-quality material with high-conductivity, the use of a nitrogen atmosphere can play a vital role in minimizing contamination (especially by oxygen, moisture). By creating a clean and controlled atmosphere, we can reduce the presence of impurities, and achieve a designed product with enhanced properties. Therefore, in the context of our synthesis procedure, the utilization of a nitrogen atmosphere is advantageous in minimizing contamination by reducing the presence of impurities commonly found in argon atmospheres, and ensuring the production of a highly conductive final product.
- 3. Cost considerations:** Nitrogen is generally more readily available and less expensive compared to argon. When cost is a significant factor in the synthesis procedure, opting for a nitrogen atmosphere can offer an economical advantage without compromising the overall success of the process. Choosing a nitrogen atmosphere can be a cost-effective solution, particularly in situations where the specific properties of argon are not critical to the synthesis procedure. Nitrogen can still provide the necessary inertness and thermal stability required for many high-temperature synthesis processes. By utilizing a readily available and cost-efficient nitrogen atmosphere, we can achieve the desired synthesis outcomes without incurring excessive expenses associated with using argon. In our case, considering the synthesis procedure conducted at 3000 degrees Celsius, the cost considerations are important, and employing a nitrogen atmosphere can be a more

financially viable choice. This decision can help optimize resource utilization and enable the successful synthesis of the material without compromising the overall results. The availability and lower cost of nitrogen make it a practical choice for industrial-scale synthesis procedures. By utilizing a nitrogen atmosphere, we can optimize resource utilization and minimize expenses without compromising the quality or integrity of the synthesized material. This cost advantage can be particularly beneficial in industrial applications where large-scale production is required. Therefore, the utilization of a nitrogen atmosphere in our synthesis procedure not only provides an economical advantage but also maintains the required environment for successful synthesis, thereby amplifying the possibilities for industrial applications of the developed material.

A brief description has been added in the main text:

(Page 7): "After acid-washing and heat treatment at 3000°C under a nitrogen atmosphere, the N-CNTs@MC composite was obtained (Supplementary Fig. 5). The nitrogen atmosphere was selected due to the possibility of chemical reactions, reduced contamination, and cost-effectiveness in this process."

Comment–2: In CV: beyond 4.2 V especially during anodic sweep, a sudden increase in current is more prominent, which might be indicating electrolyte oxidation. please explain. Additionally does author has any other data of electrolyte decomposition didn't happen at that higher voltages, such as XPS of used electrode etc.

Response to Comment–2:

We thank the reviewer for careful reviewing and raising this question. Indeed, the current increase more prominent beyond 4.2 V in CV for the full cell. But it is less likely to be resulted from the electrolyte oxidation. As shown in **Fig. R17 (Supplementary Fig. 1)**, the cathode exhibits normal capacitance curve beyond 4.2 V, indicating the electrolyte are not decomposed beyond 4.2 V. We used the same electrolyte for half cell and full cell, which means the electrolyte is stable beyond 4.2 V in the full cell.

Fig. R17. *a* Schematic of the mechanism for potassium storage in PIBHCD. *b* The relative shapes of CV curves for full cell. (Supplementary Fig. 1)

The reason for the significant current increase beyond 4.2 V in CV for the full cell is attributed to the capacitive current. As shown in **Fig. R17 (Supplementary Fig. 1a)**, the current increase dramatically at high voltage for the cathode and at low voltage for anode, respectively. As a result, the full cell using this anode and cathode could show fast increase in the current at high voltage.

Additionally, we performed measurements to confirm that electrolyte decomposition did not occur. X-ray Photoelectron Spectroscopy (XPS) analysis was conducted on the cathode electrode before and after 20 cycles from 0 to 4.8 V (**Fig. R18**). The presence of peaks corresponding to C-O and C=O bonds was observed in the cathode sample before cycling. However, after 20 cycles, no additional peaks were detected in the cathode electrode, indicating the absence of decomposition of the Dimethyl Carbonate (DMC) or Ethyl Methyl Carbonate (EMC) components. Furthermore, in the F 1s curve, the presence of PF₆⁻ was observed, and no new peaks appeared after 20 cycles, indicating the PF₆⁻ is stable in this voltage window.

A compelling and convincing piece of evidence can be observed through the use of a pouch cell. Normally, if the electrolyte starts to decompose, the pouch cell will exhibit bulging or swelling after a few cycles, and the electrolyte may become depleted or dry following long-term cycling. This observation serves as an intuitive and reliable indicator of electrolyte decomposition. In our work, we evaluated the 1 Ah pouch cell after 100 cycles. As depicted in **Fig. R19**, no noticeable changes in the appearance of the pouch cell were observed after 100 cycles. Furthermore, during the disassembly of the pouch cell, it was found that the electrolyte

remained at an adequate level. In comparison, when the pouch cell cycled for 2 cycles between 0 V-5.0 V, noticeable swelling of the cell was observed, suggesting some gases formed due to the decomposition of electrolyte. These findings strongly suggest that the electrolyte does not undergo decomposition within the tested voltage window.

Fig. R18. The XPS analysis of the cathode electrode before and after 20cycles: **a.** O 1s curves, **b** F 1s curves of cathode before and after 20cycles. (Supplementary Fig. 16)

Fig. R19. **a** The photograph of a pouch cell after 100 cycles (4.8 V). **b** The photograph of the disassembled pouch cell after 100 cycles (4.8 V). **c** The photograph of the disassembled pouch cell after 2 cycles (5 V).

A brief description has been added in the main text:

(page 17): X-ray Photoelectron Spectroscopy (XPS) analysis was conducted on the cathode electrodes before and after 20 cycles from 0 to 4.8 V (Supplementary Fig. 16). The presence of peaks corresponding to C-O and C=O bonds was observed in the cathode sample before cycling. However, after 20 cycles, no additional peaks were detected in the cathode electrode,

indicating the absence of decomposition of the Dimethyl Carbonate (DMC) or Ethyl Methyl Carbonate (EMC) components. Furthermore, in the F 1s curve, the presence of PF₆ was observed, and no new peaks appeared after 20 cycles, indicating the PF₆ is stable in this voltage window.

Comment–3. Please explain the method of calculating capacitive vs diffusive current elaborately in supplementary. also please indicate coefficients value.

Response to Comment–3:

We thank the reviewer for careful reviewing and providing this valuable comment. In the supplementary section, we provide a detailed explanation of the method used to calculate the capacitive and diffusive current contributions in our system. We also include the coefficient values used in the calculations. Here is the elaboration:

To determine the capacitive and diffusive current components, we employed a widely used method based on the differentiation of the galvanostatic charge-discharge curves. The total current (I) can be expressed as the sum of capacitive current (i_c) and diffusive current (i_d) contributions: $i = i_c + i_d$.

The peak current observed during electrochemical measurements for potassium-ion storage can be attributed to two main processes: capacitive current and diffusive current. The capacitive current arises from charge transfer processes occurring at the electrode-electrolyte interface. This current is typically associated with the adsorption and desorption of ions onto the electrode surface. The charge storage in this case is predominantly governed by the electrostatic interaction between the electrode and the electrolyte ions.

The relationship between the peak current density (i) and the sweep rate (v) could be described as the following equations [*Nature Mater.* 12(6), 518–522 (2013)]:

$$i = av^b \quad (1)$$

$$\log(i) = b \cdot \log(v) + \log(a) \quad (2),$$

where v is the scan rate, i is the peak current, a and b are constants.

The b value could be determined by the slope of linear fitted log(i)-log(v) curves. The values of a and b in Equations (1) and (2) are determined through experimental data analysis. These equations are commonly used to fit the relationship between the scan rate (v) and the peak current (i) observed during electrochemical measurements for potassium-ion storage.

In Equation (2), the logarithmic transformation of Equation (1) is applied to linearize the relationship between log(i) and log(v). This linear form allows for a more straightforward

analysis of the data and provides insights into the underlying mechanism of potassium-ion storage.

The b value in Equation (2) is crucial for determining the storage mechanism. When the b value approaches 0.5, it suggests an ionic diffusion-controlled behavior. This means that the potassium ions diffuse through the electrode material during the charging and discharging processes. This diffusion can occur in the bulk of the material or through pathways such as grain boundaries or defects. The b value close to 0.5 indicates that the rate of ion diffusion significantly influences the overall charge transfer process. On the other hand, the b value near 1.0 indicates a surface-controlled capacitance behavior. This suggests that the charge storage mechanism primarily occurs at the electrode-electrolyte interface. The capacitance behavior is related to the electrochemical double-layer capacitance, where the charge is stored as a result of the electrostatic interaction between the electrode surface and the electrolyte ions. In this case, the specific surface area and the electrode/electrolyte interface play crucial roles in determining the charge storage capacity.

It's important to note that these interpretations are general trends and can vary depending on the specific system and experimental conditions. The values of a and b should be analyzed in conjunction with other characterization techniques and corroborating evidence to gain a comprehensive understanding of the potassium-ion storage mechanism in a particular electrode material.

The quantitative calculation of capacitive contribution ratio during the potassium-ion storage process was conducted by the following equation [*Nature Mater.* 16(4), 454–460 (2017)]:

$$i(V) = k_1 v^{1/2} + k_2 v \quad (3)$$

$$i/v^{1/2} = k_1 + k_2 v^{1/2} \quad (4)$$

where $i(V)$ is the current density at a certain potential, v is the scan rate, k_1 and k_2 are constants at a certain potential. The values of k_1 and k_2 can be calculated through Eq. (3) and (4), which correspond to the slope and intercept of the fitted lines of $v^{1/2}$ versus $i/v^{1/2}$ plots.

The coefficient value (b -value) in our case was calculated to be 0.92, which indicates the intercalation of potassium ions tends to display a capacitive behavior after the expanding of interlayer spacing in MnO@CNTs, enabling an excellent rate performance and high capacity.

Fig. R 20. *Log(i)–log(v) curves for b-value determination (Supplementary Fig. 17)*

In the supplementary section, we provide detailed information on the specific mathematical model employed, the coefficients used, and how they were determined. This allows for a comprehensive understanding of the calculations involved in separating the capacitive and diffusive current components in our system. The updated content is as below:

“Supplementary Note 1: Calculating capacitive vs diffusive current.

To verify the contribution of pseudocapacitive behaviour, further analysis was conducted as follows. The relationship between the peak current density (i) and the sweep rate (v) could be described as the following equations¹:

$$i = av^b \quad (1)$$

$$\log(i) = b \cdot \log(v) + \log(a) \quad (2)$$

where the b value could be determined by the slope of linear fitted log(i)-log(v) curves.

Typically, when the b value is approaching 0.5, it indicates that the intercalation of potassium ion is mainly determined by the diffusion process, while the b value gets close to 1.0, the potassium storage is dominated by the pseudocapacitor behaviours, such as surface-controlled capacitive process or intercalation without phase transition.

The quantitative calculation of capacitive contribution ratio during the potassium-ion storage process was conducted by the following equation²:

$$i(V) = k_1 v^{1/2} + k_2 v \quad (3)$$

$$i/v^{1/2} = k_1 + k_2 v^{1/2} \quad (4)$$

where i(V) is the current density at a certain potential, v is the scan rate, k₁ and k₂ are constants at a certain potential. The values of k₁ and k₂ can be calculated through Eq. (3) and (4), which correspond to the slope and intercept of the fitted lines of v^{1/2} versus i/v^{1/2} plots.”

Comment-4: In case of anode, use of MnO QDs is not explained adequately. In discussion, the authors described the entire phenomenon mainly happening due to capacitive type process for which defects and surface helps. In that case how QDs are helping and conductivity of QDs are questionable.

Response to Comment-4:

We thank the reviewer for careful reviewing and raising this question. In this work, the MnO QDs mainly perform two functions: enlarging the interlayers of the carbon nanotubes (CNTs) and providing pseudocapacitance to enhance the intercalation and diffusion kinetics of K ions. In the K ion hybrid capacitor, the anode suffers from slow intercalation and diffusion kinetics, mismatching with the rapid kinetics for cathode. This significantly hinder the practical application of this kind of energy storage. Therefore, the enlarged spacing and pseudocapacitance resulted by the MnO QDs is vital to the optimization of the anodes to match the cathode well.

Fig.R13. *a* Schematic diagram, *b* TEM, and *c* enlarged TEM images of MnO@CNTs; *d* High resolution TEM image on the wall of MnO@CNTs; *e* Atomic image of the MnO quantum dots; *f* TEM image of MnO@CNTs with clear diffraction fringes of carbon layer and *g* corresponding carbon layer spacing. (Supplementary Fig. 7)

To be specific, the embedded MnO quantum dots can enlarge the interlayer space of carbon layers, which can significantly enhance the intercalation kinetics to improve the rate performance. In addition, the introduction of MnO can provide enhanced pseudocapacitance at the same time to further improve the whole kinetics, enable better compatibility with the cathode. Firstly, the MnO quantum dots and the enlarged spacing can be observed from TEM measurement in **Fig. R14 (Supplementary Fig. 7)**. Furthermore, the XRD analysis (**Fig. R14**) provides additional evidence supporting these findings. The DFT calculation has been conducted to analyse the adsorption and migration behaviour of K^+ ions in MnO@CNTs (**Fig. R10**). The results showed that MnO@CNTs composites have better compatibility with K ions and can effectively promote their adsorption on electrode materials, and the K^+ ions can migrate rapidly in MnO@CNTs compared with CNTs. The pseudocapacitance contribution of MnO@CNTs composite have been investigated as well (**Fig. 3c, d**). The enhanced intercalation kinetics and pseudocapacitance can significantly improve the capacity and rate performance of CNTs. As a result, the energy-power characteristics of the MnO@CNTs film as an anode electrode for potassium ion half cells are superior to most of the reported materials (**Fig. 3e**)

Fig. R10 Adsorption energies and energy barriers for K^+ ion on graphite and MnO@graphite based on DFT calculations. (**Updated Fig. 3h**)

Fig.R14. XRD patterns of MnO@CNTs and CNTs: **a** from 22° to 30°, **b** from 30° to 80°. **c** Raman spectra of MnO@CNTs. (Updated Supplementary Fig. 14)

The discussion has been updated in the main text to clarify it:

(Page 5): “The MnO quantum dots inlaid in the CNT walls can enlarge the interlayer space of carbon layers, which can significantly enhance the intercalation kinetics to improve the rate performance^{19, 20}. In addition, the introduction of MnO can provide enhanced pseudocapacitance at the same time to further improve the whole kinetics, enable better compatibility with the cathode. The combination between the matched cathode and anode can achieve high capacity and high-rate performance simultaneously.”

Comment–5. Raman data has been included which is appreciated but the explanation regarding same is inadequate. Please explain about the composite with Raman spectra data.

Response to Comment–5:

We thank the reviewer for proving this valuable suggestion. The Raman analysis of MnO@CNTs is presented in **Fig.R14c (Supplementary Fig. 14c)**. The spectrum reveals distinct bands, with the MnO band appearing at 640 cm⁻¹, the D band at 1336 cm⁻¹ and G band at 1569 cm⁻¹, respectively. The D band is attributed to defect edges and lattice distortion, while the G band signifies the crystalline graphitic structure. The intensity ratio of I_D/I_G serves as a measure of carbon defects. In the case of MnO@CNTs, the I_D/I_G ratio is determined to be 1.04, indicating a relatively high level of defects in the MnO@CNTs structure. Compared with the Raman of MWCNTs²⁷, the broader full width at half maximum (FWHM) of band G and the absence of the 2D peak at 2696 cm⁻¹ further suggest the enhance defects in the MnO@CNTs. These defects primarily arise from the incorporated MnO (as depicted in **Fig. R10**) and some N doping (as shown in **Fig. R21**). The Raman analysis aligns with the findings from XRD and XPS analyses, collectively reinforcing the consistency of the results.

Fig.R21. XPS survey of MnO@CNTs. (Supplementary Fig. 15)

The updated information for Raman in the main text is as below:

“The Raman analysis of MnO@CNTs is presented in **Supplementary Fig. 14c**. The XRD curve of MnO@CNTs exhibit prominent MnO peaks. When compared to the CNTs, the MnO@CNTs display a broad peak between 24-26 °C, which is attributable to defects resulting from the embedded MnO in the carbon layer and nitrogen doping. Moreover, the peak of MnO@CNTs exhibits a slight leftward shift compared to the CNTs, indicating an enlarged interlayer spacing, consistent with the structural analysis conducted via TEM. The spectrum reveals distinct bands, with the MnO band appearing at 640 cm^{-1} , the D band at 1336 cm^{-1} and G band at 1569 cm^{-1} , respectively. The D band is attributed to defect edges and lattice distortion, while the G band signifies the crystalline graphitic structure. The intensity ratio of I_D/I_G serves as a measure of carbon defects. In the case of MnO@CNTs, the I_D/I_G ratio is determined to be 1.04, indicating a relatively high level of defects in the MnO@CNTs structure. Compared with the Raman of MWCNTs²⁷, the broader full width at half maximum (FWHM) of band G and the absence of the 2D peak at 2696 cm^{-1} further suggest the enhance defects in the MnO@CNTs. These defects primarily arise from the incorporated MnO (as depicted in **Fig. 3h**) and some N doping (as shown in **Supplementary Fig. 15**). The Raman analysis aligns with the findings from XRD and XPS analyses, collectively reinforcing the consistency of the results.”

Comment–6: When using the GO the wt% or V% can be inserted in main result and discussion once for better readability

Response to Comments–6:

We thank the reviewer for proving this valuable suggestion. We have taken the feedback into careful consideration and made the necessary revisions. Specifically, we have now included the mention of incorporating 10 wt% in the main content when discussing the process of preparing the free-standing electrode. These revisions have been appropriately highlighted within the main content.

“By mixing with a small fraction of rGO (10 wt%), the N-CNTs@MC can be assembled into free-standing carbon papers which can be folded into different shapes (Fig. 2b).”

“The as-obtained MnO@CNTs could be shaped into carbon paper by mixing with a small proportion of rGO (10 wt%) (Fig. 3b), showing good flexibility and good mechanical property.

Regarding the abbreviation of the two samples, namely N-CNTs@MC-rGO and MnO@CNTs-rGO, we did not include the 10 wt% designation. This decision was made based on the fact that the proportion of reduced graphene oxide (rGO) in both samples is identical and it has been mentioned in the main context.

Comment–7: Authors are kindly requested to check for language once again as we could find various grammatic errors and spelling mistakes such as “Founding for funding” and “faction for fraction” etc.

Response to Comment–7:

We thank the reviewer for the thoughtful reviewing. We have carefully double checked the content and revised these errors, which are highlighted in the manuscript.

Comment-8: Overall, the approach reported here is novel enough and results stated are satisfactory with some logical gap. The article may be considered for publication only after major revision filling those logical gaps.

Response to Comments–8:

We thank the reviewer for the thoughtful reviewing. We appreciate your positive feedback on the novelty of our approach and the satisfactory results presented. We acknowledge your concerns regarding the presence of some logical gaps in the manuscript. We have carefully

incorporated your suggestions and comments in the revision process to effectively address these logical gaps, thereby enhancing the integrity and coherence of our work. We believe that the revised version and additional information we have supplied significantly bolsters the quality of our manuscript.

REVIEWERS' COMMENTS

Reviewer #1 (Remarks to the Author):

This manuscript represents a quality contribution in the field of electrochemical energy storage systems on a fairly new technology, K-ion hybrid capacitor. The authors took into account the comments of the reviewers, which improved the quality of the manuscript. In addition, the authors were able to answer the questions asked in order to justify their point. Thanks to the modifications and the answers given, I recommend the acceptance of this manuscript.

Reviewer #2 (Remarks to the Author):

The authors have addressed all the concerns raised by the reviewers. In addition, they have also added new data which supports the work and the conclusion. I recommend acceptance of the article in its present form.

Reviewer #3 (Remarks to the Author):

This revised version is cleared all questions. I accept all in this state.